# Observation of topological hydrogen-bonding domains in physical hydrogel for excellent self-healing and elasticity

Shaoning Zhang[1,2,3], Dayong Ren[3], Qiaoyu Zhao[4], Min Peng[2], Xia Wang[2], Zhitao Zhang [1], Wei Liu [2] ✉ & Fuqiang Huang [1] ✉

Physical hydrogels, three-dimensional polymer networks with reversible cross-linking, have been widely used in many developments throughout the history of mankind. However, physical hydrogels face significant challenges in applications due to wound rupture and low elasticity. Some self-heal wounds with strong ionic bond throughout the network but struggle to immediately recover during cyclic operation. In light of this, a strategy that achieves both self-healing and elasticity has been developed through the construction of topological hydrogen-bonding domains. These domains are formed by entangled button-knot nanoscale colloids of polyacrylic-acid (PAA) with an ultra-high molecular weight up to 240,000, further guiding the polymerization of polyacrylamide to reinforce the hydrogel network. The key for such colloids is the self-assembly of PAA fibers, approximately 4 nm in diameter, and the interconnecting PAA colloids possess high strength, simultaneously acting as elastic scaffold and reversibly cross-linking near wounds. The hydrogel completely recovers mechanical properties within 5 h at room temperature and consistently maintains >85% toughness in cyclic loading. After swelling, the hydrogel has 96.1 $wt\%$ of water content and zero residual strain during cycling. Such physical hydrogel not only provides a model system for the microstructural engineering of hydrogels but also broadens the scope of potential applications.

Hydrogel, a polymer network swollen with water, has attracted great interest in the fields of wearable devices[1,2], human-machine interaction[3,4] and biorobotics[5,6] due to its biocompatibility and diverse functionality[7]. Self-healing and elastic hydrogels are crucial for such applications to restore from the unexpected mechanical damage and prolong their service life. Generally, self-healing capability in hydrogels is commonly achieved by introducing directional self-assembly to enable the dynamic bond reconstruction[7–15]. For instance, the polyampholytes P(NASS-co-MPTC), assembled by the ionic association of single polymer network, physically crosslinks the polymer network and enables a healing efficiency of 100% after being immersed in water for 24 h[9]. However, due to high bonding strength, the self-assembly requires considerable energy (light, thermal, etc.) and enough time to reach thermodynamically stable state[16–23]. Such reversible polymer network breaks upon successive stretch and hardly recovers timely, resulting in the deterioration of elasticity[22]. Double-

[1]State Key Lab of Metal Matrix Composites, School of Materials Science and Engineering and Zhangjiang Institute for Advanced Study, Shanghai Jiao Tong University, Shanghai, China. [2]School of Physical Science and Technology, ShanghaiTech University, Shanghai, China. [3]State Key Laboratory of High Performance Ceramics and Superfine Microstructure, Shanghai Institute of Ceramics, Chinese Academy of Sciences, Shanghai, China. [4]State Key Laboratory of Molecular Biology, Shanghai Institute of Biochemistry and Cell Biology, Center for Excellence in Molecular Cell Science, University of Chinese Academy of Sciences, Chinese Academy of Sciences, Shanghai, China. ✉e-mail: liuwei1@shanghaitech.edu.cn; huangfq@sjtu.edu.cn

network hydrogels introduce additional polymer network, designed to retain self-assembly from deterioration, yet the self-healing capability and elasticity are still far from satisfactory[24,25]. Therefore, it is necessary to design a proper reversible polymer network with recoverable mechanical dissipation to ameliorate the elasticity.

Strategies for enhancing the recoverability of polymer networks include chemical crosslink and weak hydrogen-bonding (H-bonding). The presence of chemical crosslink pinches the polymer network with immobilized knots. Until these knots break, polymer chains can slip and be thoroughly recoverable within limited strain, resulting in minimal mechanical dissipation and excellent elasticity[26–28]. Meanwhile, weak H-bonding enables recoverable mechanical dissipation during cyclic loads. Instead of strong reversible bonds in self-assembly, weak H-bonding is noncooperative and thereupon rapidly reconstruct during cyclic loads. With the chemical crosslink as scaffold, the weak H-bonding between poly-vinyl alcohol and polyacrylamide (PAM) readily breaks and reforms, resulting in nearly 100% recoverable mechanical dissipation during 5000 cyclic loads[29]. Although both chemical crosslink and weak H-bonding endows hydrogel with high elasticity, neither constructs a reversible network for self-healing capability. Therefore, a strategy to achieve both recoverable mechanical dissipation and reversible network is urgent for designing elastic, self-healing hydrogels.

The topology of a polymer network stems from the assembly of polymer chains due to intermolecular interactions and steric hinderance and plays important roles in macroscopic properties, in which multiple functionalities can be regulated[24,30]. For instance, hydrophobic assembly by NaCl solution reconstructed the polymer network from the random distribution into certain topology where discrete hydrophobic domains hold ionic polyelectrolyte chains, leading to 90% self-healing capability within 24 h and recovering the full mechanical property after resting for 4 min. Yet the high strength of cooperative H-bonding and ionic interaction between polyelectrolyte chains deteriorates the polymer network during successive loading[24]. Sun et al. introduced an interconnecting hydrophobic assembly in hydrogel so that the hydrophobic domains locked the non-covalent interaction, enabling elastic contraction of polymer chains and healability[31]. However, such hydrophobic domain was only healable in 333 K water due to limited mobility of hydrophobic assembly, restraining the further application.

Herein, we demonstrate a topological design to simultaneously achieve full self-healing capability and high elastic recovery in physical hydrogel. We selectively utilize the topology of polyacrylic acid (PAA) colloids to assemble double-network hydrogel and locally construct the reversible elastic cooperative H-bonding domain embedded in noncooperative H-bonding matrix. PAA colloids are assembled via intermolecular cooperative H-bonding in concentrated solution, and different molecular weights lead to distinctive entangled conformation at the same concentration, as elucidated via Tyndall effect, cryo-TEM, small-angle X-ray scattering (SAXS), rheology and simulation. Cooperative H-bonding domain is composed of PAA colloids and polymerized acrylamide (AM) around PAA, contributing to the reversibility of polymer network with strong H-bonding. The excess AM polymerizes aside as noncooperative matrix connecting to the cooperative domain, facilitating the rapidly recoverable polymer network with weak H-bonding. The topology of cooperative domains critically determines the macroscopic properties of hydrogels, and only interconnecting cooperative domains assembled by high-molecular-weight PAA enable hydrogel with an autonomous self-healing capability (-100% healing in 5 h at room temperature) and excellent elasticity (<6.3% residual strain, >100% stress retention, >85% reversibility) upon successive loading. After swelling, the hydrogel exhibits 0% residual strain upon cyclic loading. We believe that the proposed methodology holds significant potential for designing with multiple functionalities for various practical applications.

## Results and discussion
### Structural evolution of PAA depending on molecular weights
The hydrogel was synthesized via the free-radical polymerization of AM along the PAA colloid. PAA is a well-known water-soluble polymer with alternating carbonyl and carboxyl groups and leads to high negative charge density when the carboxyl groups dissociate. The strong negative charges on the PAA backbone induces the repulsive electrostatic screening in aqueous solution and thereupon self-assembles into colloids. Due to the intermolecular cooperative H-bonding and steric alignment, PAA of diverse chain length forms colloid with different structure (Fig. 1a)[32,33]. For example, PAA chains with molecular weights of 3000 (PAA3k) and 240,000 (PAA240k) have chain lengths of approximately 10 nm and 780 nm, respectively, based on the length of monomer (-2.18 Å) and molecular weight. Due to the negligible steric hinderance, low-m.w. PAA substantially contacts with water molecules and thereupon spreads as discrete islands. Meanwhile, high-molecular-weight PAA interacts with each other due to the significant steric hinderance, self-assembling into nanostrings. Upon critical concentration, PAA nanostrings overlap and assemble highly entangled colloids with button-knot morphology (Supplementary Fig. 1), forming button-knot cooperative domains.

Before the polymerization, partial AM penetrates into PAA colloids since amide groups form Lewis-acid-base couple with carboxyl groups. Aside from the coupled AM, excess AM dissociates around the PAA-AM colloids. With the initiator ammonium persulfate (APS) activated by heat, the coupled AM in-situ polymerizes into PAM chains and keeps the supramolecular structure with PAA, forming topological cooperative domains with cooperative H-bonding. The cooperative H-bonding consists of double H-bonding between carboxylic groups and amide groups (Supplementary Fig. 2), contributing to high bonding strength. Meanwhile, the polymerization of dissociated AM forms noncooperative H-bonding matrix interconnecting with the topological cooperative domains. The noncooperative H-bonding matrix is instead composed of noncooperative H-bonding between amide groups and thereupon easily dissociates compared to cooperative H-bonding, making cooperative domains the scaffold upon deformation. Due to the discrete distribution, the cooperative domains built by PAA3k chains only reach surrounding PAM chains after polymerization, inducing permanent cut-off damage and continuous disentanglement of the polymer network. Instead, the button-knot cooperative domains by PAA240k colloids become reversible elastic scaffold and stabilize the entire matrix, resulting in highly recoverable mechanical dissipation. Such cooperative domains also knit the mechanical damage at interface, providing the hydrogel with fully self-healing capability.

The variation of colloidal behavior leads to different macroscopic characteristics of PAA solution even at the same concentration. 25 $wt\%$ PAA240k solution exhibits typical shear-thinning effect at a shear rate of 30 s$^{-1}$ (Supplementary Fig. 3) whereas 25 $wt\%$ PAA3k solution is nearly Newtonian liquid, indicating that PAA240k forms colloidal suspension in water. Such PAA240k solution exhibits a more intense Tyndall effect, and PAA240k droplet is highly viscous compared to PAA3k droplet (Fig. 1b), indicating the presence of highly entangled PAA240k colloids. Notably PAA240k and PAA3k solutions show different colors after they dye with Coomassie brilliant blue, which turns green at the pH value < 1. The color difference results from the low dissociation degree of PAA240k chains. The carboxyl group of PAA240k is associated with each other due to the steric hinderance, leading to lower H$^+$ concentration.

The colloidal behavior is further evidenced by the Cryo-TEM observation (Fig. 1c). Due to the limited size and high association with water, low-m.w. PAA solution (PAA3k, m.w. = 3000) barely exhibits discernable morphology, while high-molecular-weight PAA solution (PAA240k, m.w. = 240,000) leads to the colloids around 7 nm distributed in ice with two to four PAA nanostrings (the radius is around

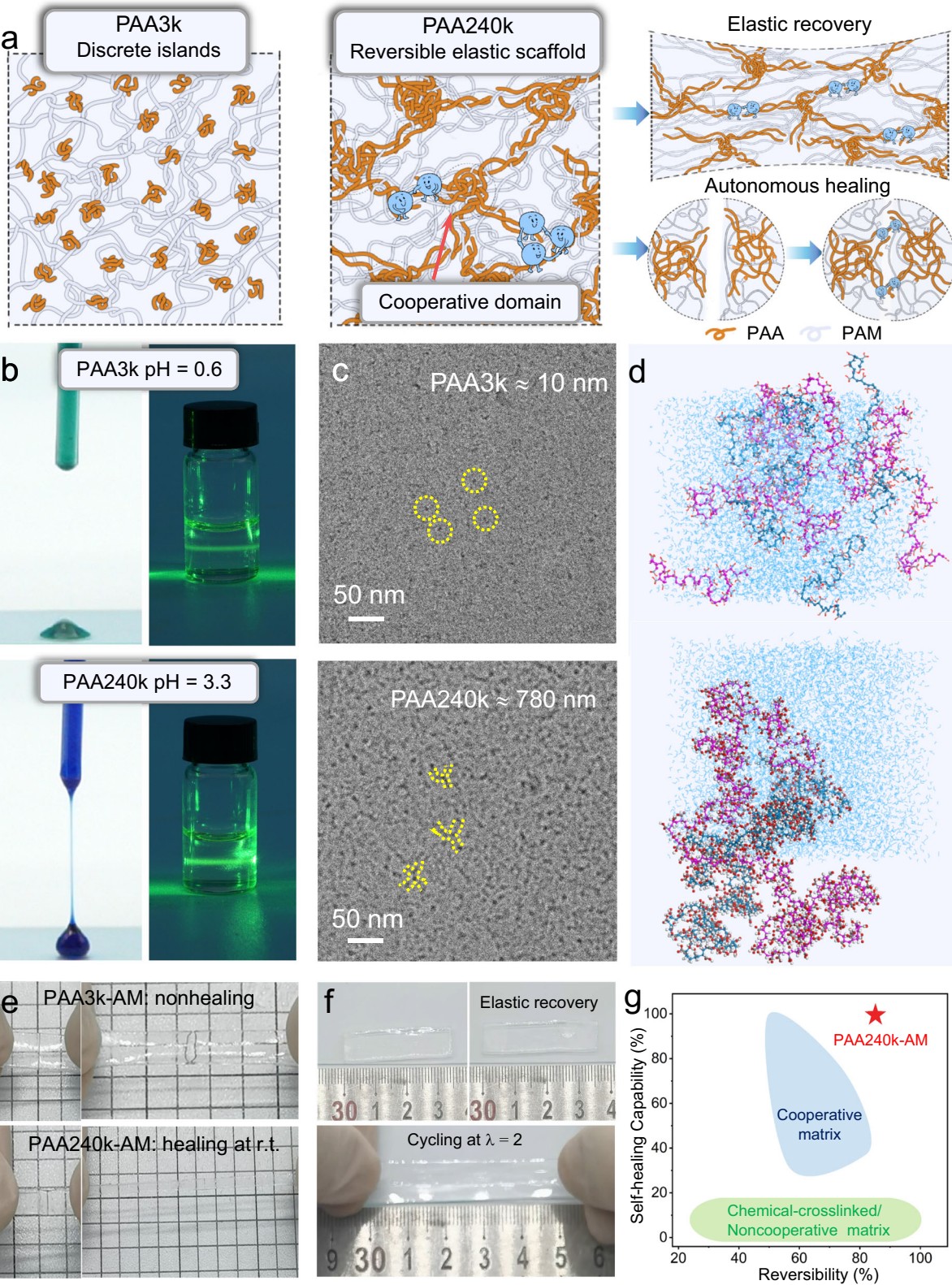

**Fig. 1 | Molecular-weight-dependent evolution of polyacrylic acid (PAA) colloids. a** Schematic of the synthesis and stretching of topological hydrogels. Low-molecular-weight PAA (orange) chains form discrete colloids and partially stabilize the surrounding polyacrylamide (PAM) (grey) chains. High-molecular-weight PAA chains form button-knot colloids and result in elastic recovery and self-healing capability. Dashed circle depicts the size of colloids. **b** Photos of PAA droplet dyed by Coomassie brilliant blue and Tyndall effect of PAA aqueous solution. The variation of droplet color results from the different pH values of PAA solution with molecular weight of 3000 and 240,000 (PAA3k and PAA240k). **c** Cryo-TEM images of 25 $wt\%$ PAA3k and PAA240k solution diluted five times. The actual concentration after blotting is much higher than the initial concentration. **d** Molecular dynamics of low-molecular-weight-PAA aqueous solution (repeating unit, r.u. = 30) and high-molecular-weight-PAA aqueous solution (r.u. = 256). Blue line depicts water molecules. Both purple line and blueviolet line represent the carbon backbone of separate PAA chains. **e** Photos of PAA3k-AM and PAA240k-AM upon deformation after self-healing for 5 h. **f** Photos of PAA240k-AM cycling at $\lambda = 2$ for 10 cycles. **g** Schematic comparison of self-healing capability and reversibility between this work and previously reported hydrogels.

2 nm). The polymer chains at high molecular weight tend to entangle at high concentration and self-assemble, thereby forming button-knot structure in water.

The evolution of colloid behavior on different molecular weights of PAA is examined via molecular dynamics (Fig. 1d and Supplementary Movie 1). Low-m.w. PAA (repeating unit, r.u. = 30) spreads in water due to its affinity to water molecules and negligible steric hindrance, and ergo seventeen short chains barely overlap during the molecular dynamic simulation. Meanwhile, high-molecular-weight PAA (r.u. = 256) self-associates due to the significant steric hindrance so that intermolecular and intramolecular H-bonding becomes dominant. Therefore, two long PAA chains entangle with each other and extend into aqueous system, leading to button-knot cooperative domains. The resultant high-molecular-weight cooperative domains can knit the mechanical damage at the interface, which grants the hydrogel with fully self-healing capability (Fig. 1e and Supplementary Movie 2). Such cooperative domains also stabilize the entire hydrogel network, resulting in highly recoverable mechanical dissipation (Fig. 1f and Supplementary Movie 3). With the hierarchical H-bonding strength and the topological distribution of cooperative domains, hydrogel manages to completely heal from the cut-off and retain the stress upon successive loading (Fig. 1g), making it prominent compared to chemically crosslinked hydrogel and other self-healing hydrogels with cooperative matrix.

## Characterization of polymerization

Before the polymerization, the precursor solution was examined via small-angle X-ray scattering (SAXS). From SAXS profiles, both PAA3k and PAA240k aqueous solution at 25 $wt\%$ experience a downturn at low-q region (Fig. 2a), which typically exists in polyanionic polymer solution where the polyanionic backbone carries negative charges and leads electrostatic screening[34,35]. When the separation between polymers is lower than electrostatic screening distance, polyanionic chains self-organize and form loosely ordered structure to minimize such interaction, forming colloids and resulting in a peak at low q region around $q_0 = 0.04$ Å$^{-1}$ in case of PAA240k solution. Compared with PAA3k, PAA240k exhibits the preferential tendency towards ordered assembly due to the narrower characteristic peak, confirming that PAA240k self-assembles due to the repulsive electrostatic screening and steric hindrance. PAA3k is more mobile and thereupon has the assembled structure only after the addition of AM.

The colloidal structure was further evaluated by Lorentzian peak fitting to obtain the correlation length ($\xi$), which describes the spatial extent over which the electrostatic interactions between the charged groups on the polymer chain are significant[36,37]. The $\xi$ of PAA240k precursor solution is 30.3 Å (Fig. 2a and Supplementary Table 1), higher than that of PAA3k precursor ($\xi = 18.9$ Å), because the correlation length scales with the chain length in good solvent and is related to the mesh size of the polymer network in concentrated polymer solutions. Additionally, the peak position of PAA240k is significantly lower than calculated chain length of PAA240k, indicating that the peak is attributed to the structure within colloid[37]. The $\xi$ of PAA240k is close to the diameter of PAA240k nanostrings from Cryo-TEM images, in concordance with the observed mesh size of PAA240k assembly. After the addition of AM, the resultant PAA240k+AM precursor retains the characteristic peak around 0.042 Å$^{-1}$ and leads to higher $\xi$ of 32.3 Å, suggesting enhanced intermolecular interaction and extended state of PAA240k chains upon interacting with AM. Meanwhile, PAA3k solution has the fitting peak at 0.064 Å$^{-1}$ (Supplementary Table 1), which is close to the calculated chain length of PAA3k, and thereupon the peak is related to inter-colloid interactions. PAA3k+AM solution results in a characteristic peak around 0.018 Å$^{-1}$ and is fit to have a $\xi$ of 17.2 Å. The decrease in correlation length suggests the further coiled state of PAA3k chains, which may result from the steric separation by AM monomers and accompanying charge compensation.

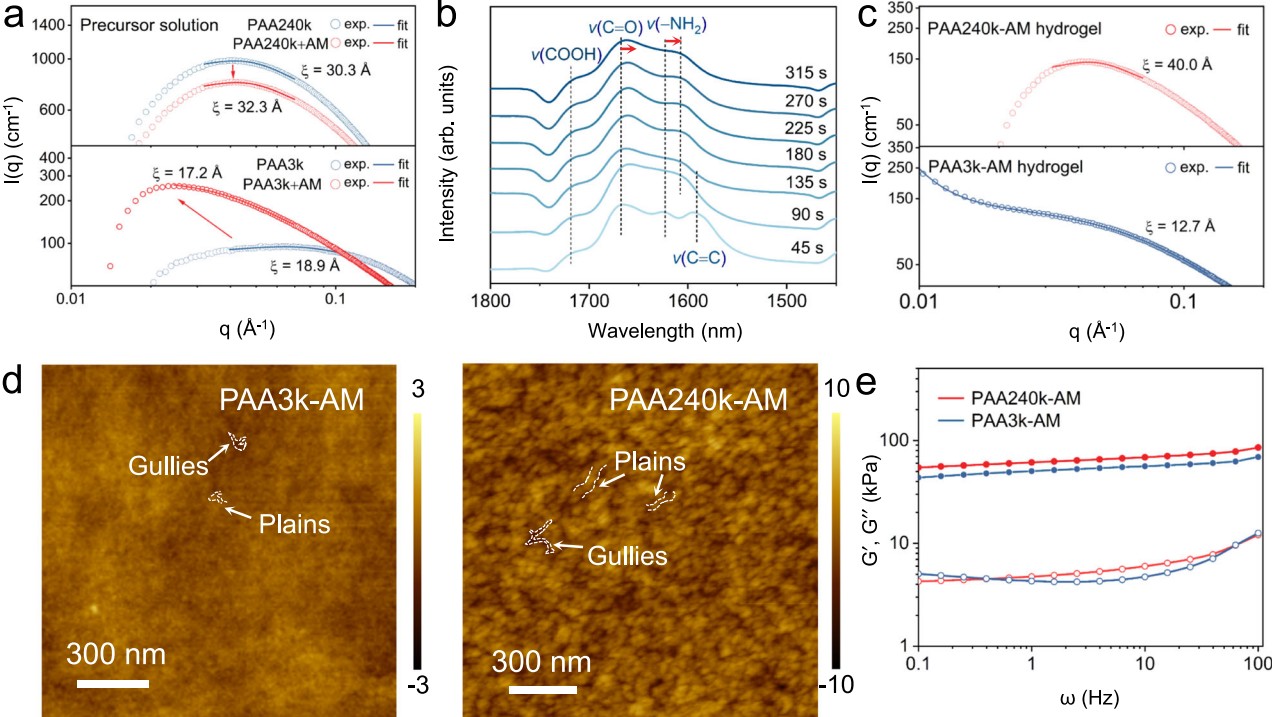

**Fig. 2 | The polymer network directed by PAA colloids. a** SAXS profiles of PAA240k, PAA240k+AM, PAA3k, and PAA3k+AM precursor solution with Lorentzian peak approximation. The fit result is correlated with correlation length ($\xi$). **b** In-situ ATR-FTIR spectra of PAA240k-AM during polymerization. Each spectra were recorded based on the polymerization time. **c** SAXS profiles of PAA-AM hydrogels with correlation length approximation and Lorentzian peak approximation. **d** AFM images of PAA3k-AM and PAA240k-AM. **e** Storage modulus ($G'$) and loss modulus ($G''$) of PAA240k-AM and PAA3k-AM hydrogels.

The polymerization of hydrogel was characterized by in-situ attenuated total reflection-Fourier transform infrared (ATR-FTIR) spectroscopy (Fig. 2b). The FTIR spectra of precursor solution exhibit four characteristic peaks of acrylamide at $1666\,cm^{-1}, 1628\,cm^{-1}$, $1501\,cm^{-1}$, and $1460\,cm^{-1}$, corresponding to the $\nu(C=O)$, $\nu(-NH2)$, $\nu(C=C)$ and $\nu(C-N)$, respectively[38]. The $\nu(COOH)$ from protonated PAA locates around $1710\,cm^{-1}$, partially overlapping with that from AM. After the polymerization of AM, the $\nu(C=C)$ vanishes and the $\nu(C-N)$ shifts to higher wavenumber. The $\nu(COOH)$ from carboxyl group shifts to lower wavenumber (Supplementary Fig. 4), suggesting the further complexation of between PAA and PAM after the polymerization. The in-situ ATR-FTIR spectrum shows that as the polymerization initiates, the AM monomer is attacked by the free-radical from initiator as the $\nu(-CH=CH_2)$ rapidly vanishes. Notably, the $\nu(C=O)$ retains at the same wavenumber until the polymerization ends, implying that the bonding environment with AM does not change along the polymerization.

SAXS was further conducted to characterize the structural information of cooperative domains of hydrogels. The down-turn phenomenon retains in PAA240k-AM hydrogel after the polymerization yet is absent in PAA3k-AM hydrogel (Fig. 2c), reflecting that only PAA240k colloids keep the interaction. The repulsive interaction between PAA3k is possibly compensated by PAM chains due to the lack of steric hinderance, leading to a minor peak around $q_0 = 0.02\,Å^{-1}$. From the gaussian size fitting (SasView, 5.0.5 Version), the PAA240k-AM hydrogel shows the similar peak position at $0.0438\,Å^{-1}$ with precursor solution (PAA240k: $q_0 = 0.0434\,Å^{-1}$, PAA240k+AM: $q_0 = 0.0434\,Å^{-1}$) (Supplementary Fig. 5 and Supplementary Table 2), again confirming that the colloidal structure is retained to form cooperative domains throughout the polymerization. The ξ of PAA240k-AM is further intensified from 32.3 Å to 40.0 Å after the polymerization (Figs. 2a and c and Supplementary Table S1), indicating the stronger intermolecular interaction within cooperative domain to sprawl in noncooperative matrix. Meanwhile, the ξ of PAA3k-AM decreased from 17.2 Å to 12.7 Å after the polymerization (Supplementary Table 3), indicating that discrete islands further contract and reach limited PAM chains[39].

PAA-AM hydrogels demonstrates different polymer network due to the directed polymerization around different PAA colloids[40]. Field-emission scanning electronic microscope (FESEM) images show that PAA240k-AM has uniform morphology at micron level, which is different from typical hydrogels with mesopores or micropores[10,41,42], while PAA3k-AM leads to bumpy surface (Supplementary Fig. 6). At submicron scale, atomic force microscope (AFM) images show that both hydrogels exhibit the mixed morphology of plains and gullies where plains are polymer regions and gullies are the residual of sublimed ice (Fig. 2d). PAA3k-AM has dense yet tiny plains at nanoscale, and PAA240k-AM leads to interconnecting and stout plains, again indicating that PAA240k-AM has continuous PAA scaffold, but PAA3k-AM has discrete domains.

The viscoelastic response was examined to investigate the stability of the polymer network (Fig. 2e)[43]. The storage modulus ($G'$) is significantly higher than loss modulus ($G''$) at all frequencies for all hydrogels, indicating their elastic and solid-like nature. Specially, the $G'$ of PAA240k-AM reaches 61 kPa at $\omega = 1\,rad\,s^{-1}$, even higher than the value of reported chemically crosslinked PAM hydrogel[44,45], and is increasing across the frequency sweep. The increase of stiffness at higher frequencies indicates the entangled and non-permanent nature of polymer network, which is observed in both polymer melts and gels. The high $G'$ signifies the large stored deformation energy in hydrogel network and would act as the driving force to reform the hydrogel structure to original shape. On the other hand, the $G''$ of the topological hydrogel is significantly lower than that of the copolymer hydrogel (Supplementary Fig. 7), indicating less viscous contribution during deformation. The stable $G'$ and $G''$ of the topological hydrogels stem from the rapid construction of noncooperative H-bonding and

contribute to the elastic recovery upon stretching. The loss factor (tan δ), defined as the ratio of $G''$ to $G'$, describes the proportions of elastic and viscous behavior. Due to the benefits of cooperative domains, the tan δ of PAA240k-AM is much less frequency-sensitive compared to P(AA-co-AM) and PAA3k-AM before $\omega = 10\,Hz$. It remains below 0.1 until $\omega = 40\,Hz$ (Supplementary Fig. 8), demonstrating behavior close to ideal elasticity (tan δ = 0)[46]. The downturn of tan δ in case of P(AA-co-AM) and PAA3k-AM indicates that both hydrogel networks relax at low frequency whilst PAA240k-AM behaves like crosslinked hydrogel.

## The self-healing and mechanical performance based on molecular weights

The topological cooperative domains, built by long-chain colloids and cooperative H-bonding, grants the PAA240k-AM hydrogel with self-healing ability. When the hydrogel is cut, the physical bonding network is separated into two parts. During the self-healing process, the nanofibers of PAA colloids diffuse through the wound interface and knit the interface with supramolecular complex. Polymers then re-entangle with each other and thereupon recover the wound. PAA240k-AM is cut and placed at room temperature in a humid atmosphere to observe the self-healing process directly under the industrial digital microscope (Fig. 3a). Initially the gap with width of 60 μm is observed, then progressively dimmers with the increasing healing time and completely vanishes after 12 h. By attaching each other, several separate hydrogel blocks also heal together in humid condition and are able to bend and stretch without damage (Fig. 3b). Besides, the mechanical property of hydrogel after cutting off is restored with the respect of healing time. PAA240k-AM shows humidity-dependent stretch fracture behavior and possesses the stretchability of 12 and maximum stress of 110 kPa at 60% humidity (Supplementary Fig. 9), while PAA3k-AM and P(AA-co-AM) only demonstrate limited stretchability and breakage stress (Supplementary Fig. 10). For PAA240k-AM, 50% of the break elongation and 60% of the original break strength are recovered after 3-h healing, and the mechanical property is utterly restored after 5-h healing (Fig. 3c, Supplementary Fig. 11 and Supplementary Movie 4). At 333 K, the self-healing efficiency of PAA240k-AM is further improved, completely restoring mechanical behavior in one hour (Supplementary Fig. 12). In comparison, short PAA chains in PAA3k-AM and loose cooperative H-bonding in P(AA-co-AM) fail to knit the surrounding polymer chains (Fig. 3c and Supplementary Fig. 13), resulting in limited self-healing capability (Fig. 3d). The elastic behavior of PAA240k-AM is also recovered by conducting successive stretch cycles (Fig. 3e).

In case of mechano-stability of PAA240k-AM, the loading curve increases with higher stretching rates due to the different amount of broken sacrificial bond and the higher interchain friction, which is a typical mechanical behavior of entangled polymer network (Supplementary Fig. 14)[27]. Successive stretch cycles at different humidities shows that PAA240k-AM is capable of bouncing back, dissipating energy along the loading cycle and achieving the same stress as that of the first cycle above 60% humidity (Fig. 3f and Supplementary Fig. 15). In contrast, PAA3k-AM and P(AA-co-AM) fail to reach the same stress during successive cycling. Notably, the second loading curves of all hydrogels overpass the previous unloading curve, reflecting that the H-bonding network reforms during successive loading.

The elasticity of hydrogel is comprehensively evaluated by multiple parameters such as reversibility and residual strain. The reversibility is defined as the ratio of the toughness from the corresponding loading curve to that from the first loading curve, $U_n/U_1$ (Supplementary Fig. 16)[47]. In case of cyclic loads at $\lambda = 2$, PAA240k-AM retains higher reversibility than those of PAA3k-AM and P(AA-co-AM), reflecting the robustness of button-knot PAA240k colloids in stabilizing surrounding noncooperative matrix (Fig. 3g). With the reversibility

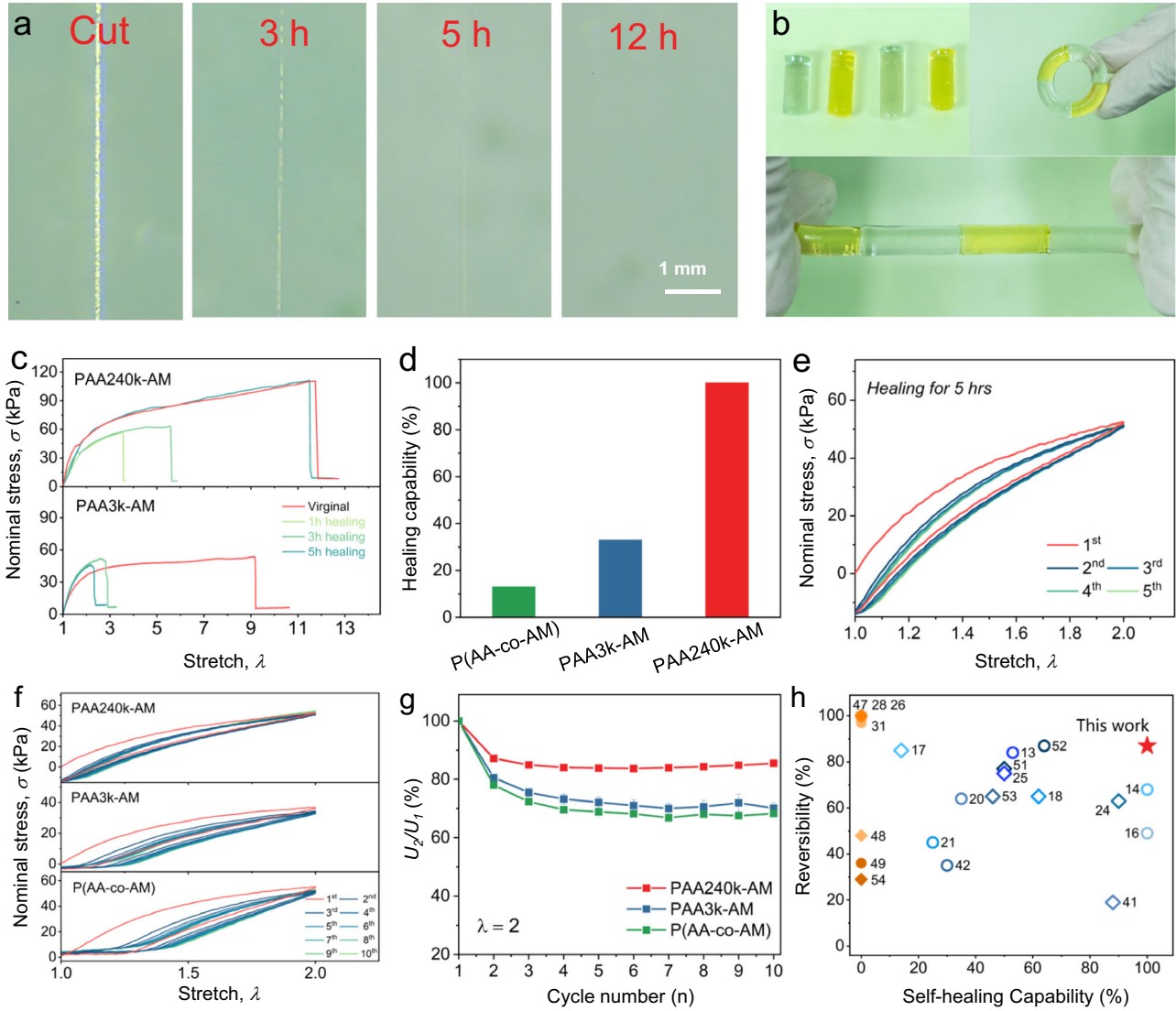

**Fig. 3 | Self-healing and mechanical performance of PAA-AM hydrogels. a** Self-healing process of cut PAA240k-AM observed by a digital microscope. **b** Photos of PAA240k-AM blocks healing towards each other. **c** Stress-stretch curves of PAA240k-AM and PAA3k-AM after different healing time. The stretch rate is 2 min⁻¹. **d** Healing capabilities of PAA-AM hydrogels after 5 h. The healing capability is calculated based on the recovered extensivity. **e** Successive stress-stretch curves of PAA240k-AM at $\lambda = 2$ after healing for 5 h. The stretch rate is 3 min⁻¹. **f** Successive stress-stretch curves of PAA240k-AM, PAA3k-AM, and P(AA-co-AM) at stretch ($\lambda$) = 2 for 10 cycles. The stretch rate is 3 min⁻¹. **g** The reversibility of PAA-AM hydrogels upon 10 cycles. Error bars belong to the standard deviation. **h** Comparison of self-healing efficiency and toughness retention between this work and previously reported hydrogels. Solid symbols belong to non-self-healing hydrogels, and hollow symbols are subtractive self-healing hydrogels.

(>85%) and self-healing efficiency (~100%), the performance of PAA240k-AM surpasses that of other reported hydrogels (Fig. 3h, Supplementary Table 6)[13,14,16,17,19,20,24–26,28,41,42,47–54].

The residual strain is defined as the unrecoverable deformation from cyclic loads and examined as the strain when slope of unloading curve abruptly changes (Supplementary Fig. 17). The initial residual strain of PAA240k-AM after the first cycle is 0.83% (Supplementary Fig. 18), the lowest among all PAA-AM hydrogels (PAA3k-AM: 11.4%, P(AA-co-AM): 16.7%). Notably, the molecular weight of PAA chains significantly affect the stability of resultant polymer network. The residual strains of all cyclic loading curves are tracked, and PAA240k-AM keeps the residual strain below 6.3% throughout 10 cycles while those of PAA3k-AM and P(AA-co-AM) soar to 26% and 34%, respectively, indicating that long-chain PAA as an interconnecting scaffold holds the surrounding PAM matrix from significant plastic deformation. Therefore, even with the same mass ratio, the topology of polymer network plays a significant role in determining elasticity, and the cooperative domain based on button-knot PAA colloids effectively stabilize the polymer network.

## The swelling behavior depending on molecular weights

Since the association degree of carboxyl group is dependent on the molecular weights of PAA chains, the as-formed PAA-AM hydrogels exhibit the variation on swelling behavior. Unlike typical physical hydrogel, PAA240k-AM swelled in all dimensions and retained the structural integrity even after 48 h (Fig. 4a), while PAA3k-AM rapidly dissolved in water and lost the initial shape after 6 h. PAA240k-AM can stabilize at the swelling ratio of 10.5 after 12 h due to high association degree and entangled polymer network (Fig. 4b), and the water content after swelling is 96.1 *wt%*. In contrast, P(AA-co-AM) and PAA3k-AM continuously swell and lose integrity after 3 h and 8 h, respectively (Fig. 4b and Supplementary Fig. 19). Absent from entangled self-assembly, the highly hydrophilic nature of acrylic acid and arylamide leads full dissociation of polymer network. SAXS pattern shows that

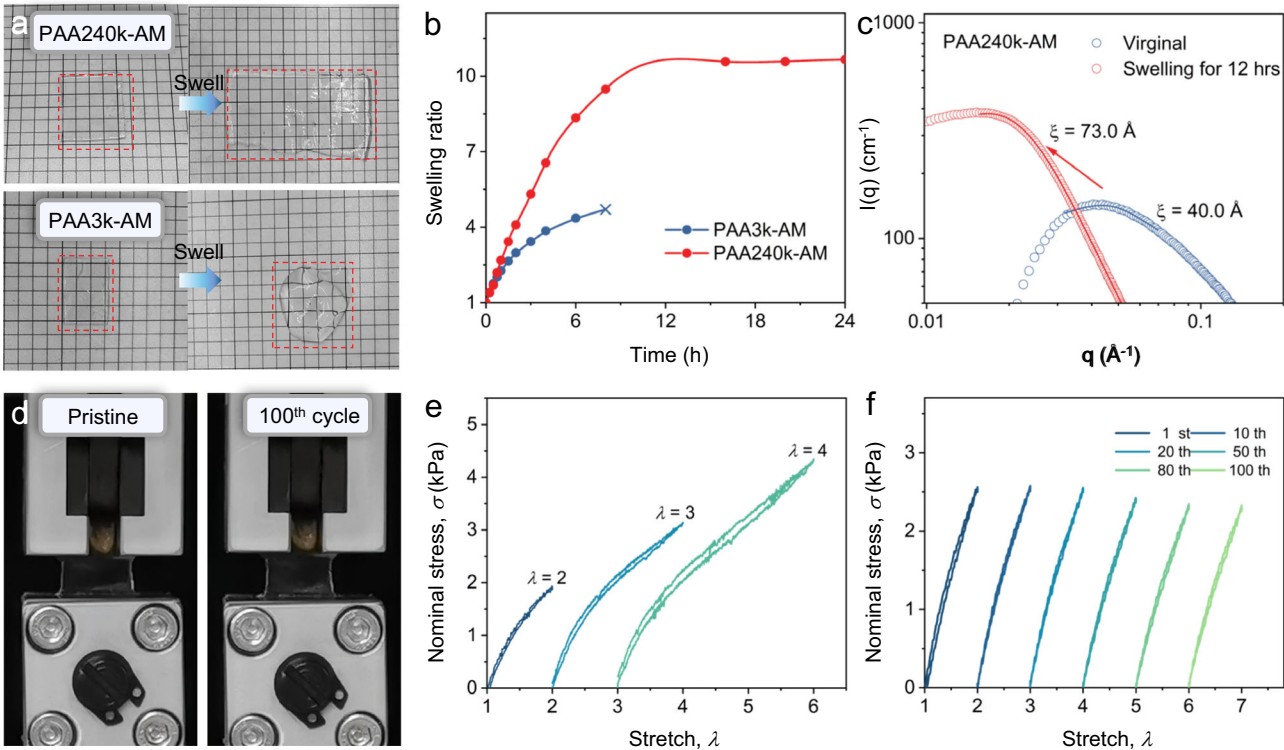

**Fig. 4 | Swelling behavior of PAA-AM hydrogels. a** Photos of PAA240k-AM after swelling for 48 h and PAA3k-AM after 6 h. Red lines circle out the hydrogel throughout swelling. **b** Swelling evolution of PAA240k-AM and PAA3k-AM across time. **c** SAXS profiles of PAA240k-AM before and after swelling with Lorentzian peak approximation. **d** Photos of swelled PAA240k-AM throughout the cycling at $\lambda = 2$. **e** Successive loading curves of swelled PAA240k-AM at $\lambda = 2$, $\lambda = 3$, and $\lambda = 4$. The stretch rate is 3 cm$^{-1}$. **f** cyclic loading curves of swelled PAA240k-AM at $\lambda = 2$ for 100 cycles.

PAA240k-AM keeps the downturn shape near low q region even after swelling for 12 h (Fig. 4c), reflecting that PAA240k chains maintain the self-assembly structure even though PAA is known to be highly associated with water. The Lorentzian peak approximation leads to the increase of ξ from 40.0 Å to 73.0 Å (Supplementary Table 5), indicating that the assembled PAA nanostrings expand upon the introduction of water and leads to highly associated yet entangled PAA nanostrings.

The assembled PAA fibers stabilize the polymer network of PAA240k-AM even after swelling. The swelled PAA240k-AM has a breaking stress of 6.4 kPa and a breaking elongation of $\lambda = 6.7$ (Supplementary Fig. 20) and exhibits no shape change upon cycling at $\lambda = 2$ for 100 loading cycles (Fig. 4d). Upon cyclic loading, the swelled PAA240k-AM exhibits no residual strain and low dissipation even at $\lambda = 4$ (Fig. 4e), demonstrating the fully recoverable polymer network. After swelling, the chain friction and non-cooperative H-bonding are minimized so that the mechanical dissipation in the virginal PAA240k-AM becomes negligible and stabilized by the entangled button-knot cooperative domains. One hundred cycles at $\lambda = 2$ also lead to zero residual strain and stable loading (Fig. 4f and Supplementary Movie 5).

### The evolution of microstructure during deformation
The mechano-stability of PAA-AM hydrogels is first evaluated from the mechanism of bonding dissociation. The average H-bonding strength in topological polymer network is investigated via a simple molecular model (Fig. 5a). The single H-bonding (4) between amide and carboxyl groups is −184 kJ/mol (Supplementary Fig. 21), higher than those other H-bondings. Considering the double H-bonding nature, the overall H-bonding strength in PAA/PAM scaffold is much stronger than the noncooperative matrix (Fig. 5b). During stretching, the noncooperative matrix easily dissociates upon deformation, and cooperative domains immobilize PAM chains as the elastic scaffold due to the variation of H-bonding strength.

Time-resolved SAXS profile tracks the evolution of polymer network during cyclic loading. The characteristic peak of PAA3k-AM around 0.04 Å$^{-1}$ gradually deteriorates (Fig. 5c), and the corresponding characteristic ring weakens during deformation and almost disappears after 10 cycles (Supplementary Fig. 22), reflecting the inter-colloid ordering continuously decreases. Notably, the ξ of PAA3k-AM remains unchanged throughout cyclic loading (Supplementary Fig. 23 and Table S3), suggesting that PAA3k remains coiled state and is weakly correlated to the deformation. In contrast, PAA240k-AM shows that the colloidal peak rapidly stabilizes around 0.034 Å$^{-1}$ after the second cycle (Fig. 5d). The ξ of PAA240k-AM decreases from 40.0 Å to 30.3 Å and stabilizes after the second cycle (Supplementary Fig. 23 and Supplementary Table 4), conforming with the trend of toughness. Meanwhile, the ξ of PAA3k-AM continuously decreases after the second cycle, reflecting that cooperative domains within PAA3k-AM substantially contact. The corresponding 2D SAXS patterns substantially exhibit significant isotropy along 10 cycles, again indicating the excellent mechano-stability of colloidal structure under cyclic loads (Supplementary Fig. 22B) and thereby confirming that button-knot colloids manage to stabilize the polymer network. After swelling, the characteristic peak of PAA240k-AM barely shifts throughout the cyclic loads, and the ξ of swelled PAA240k-AM increases to 77 Å and stabilizes (Fig. 5e, Supplementary Fig. 23 and Supplementary Table 5), indicating that the stability of polymer network is further improved when the chain friction is minimized. Therefore, due to the high bonding strength of cooperative H-bonding and entangled nature, the button-knot PAA is stable during the cyclic deformation and holds the surrounding PAM matrix (Fig. 5f). The noncooperative PAM matrix easily dissociates upon deformation, and button-knot PAA immobilizes PAM chains as the cooperative domain, leading to elastic recovery of hydrogel.

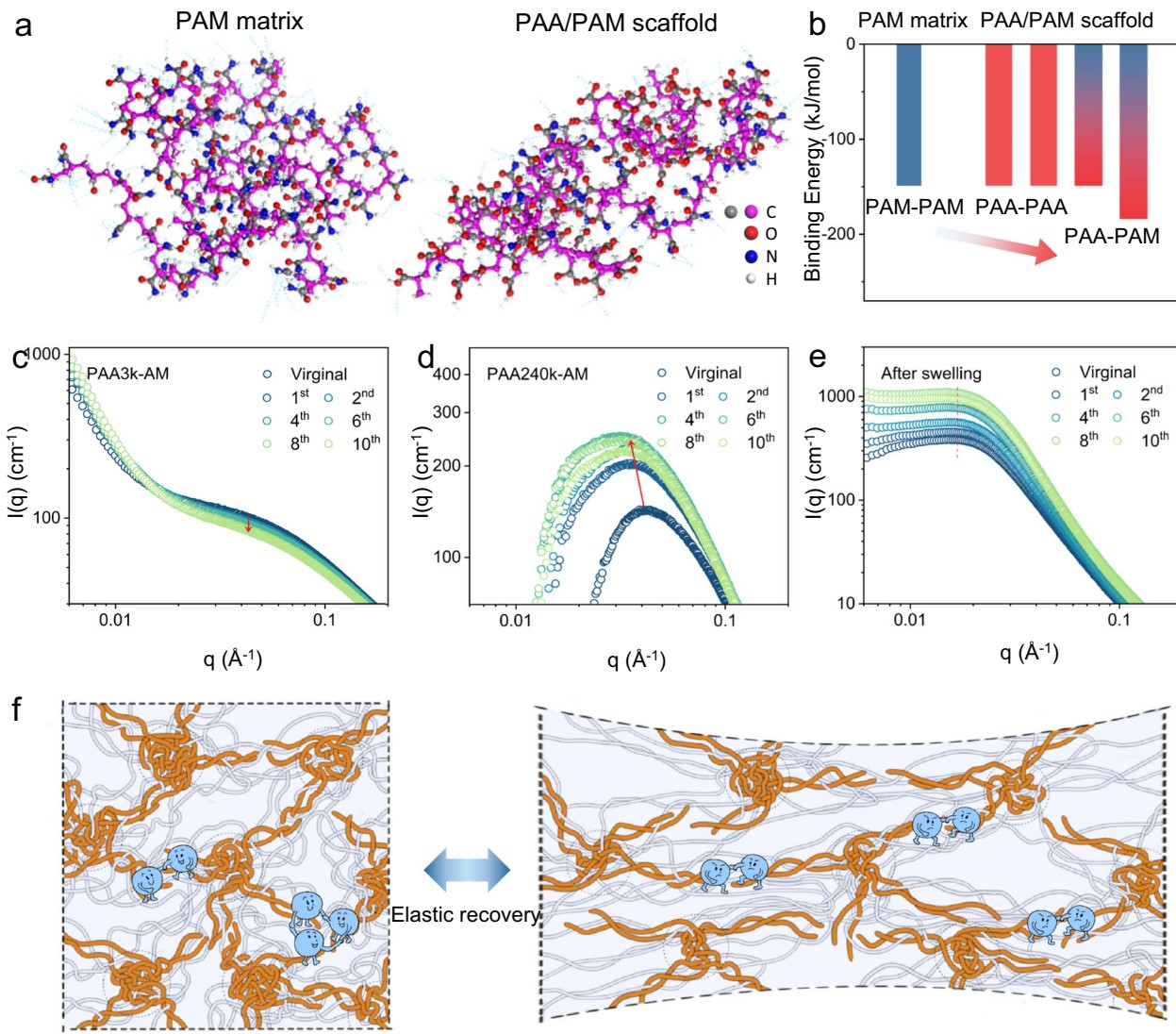

**Fig. 5 | Structural evolution of PAA-AM hydrogels. a** The molecular models of PAM matrix and PAA/PAM scaffold. **b** The binding energy of hydrogen bonding within PAM matrix and PAA/PAM colloids based on the theoretical simulation. **c**–**e** Time-resolved SAXS profiles of **c** PAA3k-AM, **d** PAA240k-AM and **e** swelled PAA240k-AM for ten successive cycles. **f** Schematic of stretching of topological hydrogel. High-molecular-weight PAA chains form button-knot colloids and result in elastic recovery.

Based on the structural evolution of two PAA-AM hydrogels, molecular weight significantly impacts the packing conformation of PAA in multi-component concentrated solution and gel. Low-m.w. PAA exhibits coiled state and is separated by PAM network, thereby losing the interaction among PAA colloids. High-molecular-weight PAA instead are assembled as flexible nanostrings and form interconnecting scaffold, securing the self-healing network and structural stability. Therefore, the self-assembly conformation of PAA determines the reconstruction of polymer network and the structural integrity during deformation. In real-world applications, mechano-stability is critical for the robustness of devices, and autonomously self-healing capability prevents devices from unexpected damage. The hydrogel in this work provides strategy to surpass the trade-off between self-healing capability and elasticity, potentially solving the requirement of next-generation applications.

In summary, we have developed a strategy on fabricating a self-healing and elastic physical hydrogel by topological cooperative domain. Such domains were constructed based on the topology of PAA chains and directed polymerization of AM. PAA chains formed the colloidal structure by repulsive electrostatic interaction and immobilized the PAM chains by cooperative H-bonding, retaining colloidal cooperative domains within the noncooperative PAM matrix. By tuning the chain length of PAA and thereupon the structure of cooperative domains, such topological structure resulted in a perfectly reversible and highly recoverable polymer network, achieving both ~100% self-healing efficiency and excellent elasticity (>100% stress retention, >85% toughness retention) during cyclic loads. The physical hydrogel still retains with 97.5 $wt\%$ after swelling and has 0 residual strain upon successive cycling. The cooperative domain not only provides the reversible cooperative bonding for wound-healing but also anchors the interconnecting noncooperative matrix, preventing irrecoverable mechanical dissipation. When the physical hydrogel is further swelled, the hydrogel leads 0% residual strain during cyclic loads. This topological strategy would shed light on the design of self-healing yet resilient hydrogels for the practical applications such as wearable electronics, human-machine interaction and biorobotics.

## Methods
### Materials
Acrylamide (AM) was obtained from Shanghai Titan Scientific Co., Ltd. (Shanghai, China). Polyacrylic acid (PAA240k, 25 $wt\%$, $m.w. = 240,000\ g\ mol^{-1}$) was purchased from J&K Scientific. Polyacrylic

acid (PAA3k, 30 *wt.%*, *m.w.* = 3000 g mol⁻¹) was purchased from Macklin Inc. (Shanghai, China). Acrylic acid (AA) and Ammonium persulfate (APS) was obtained from Adamas-beta Co., Ltd. (Shanghai, China). All materials were used without further purification.

## Synthesis of PAA-AM hydrogels

The PAA-AM hydrogel was synthesized through free-radical polymerization. PAA240k (8 g) was first diluted with 2 ml deionized water. AM (4 g) was dissolved in above PAA240k solution, marked as solution A. The solution A was vacuumed to remove bubbles. APS (1 g) was dissolved in 10 mL water, marked as solution B. One millilitre solution B was added into solution A. The mixed solution was added into glass mold with a spacer of 1 mm and reacted at 333 K for 30 min. The resultant hydrogel was marked as PAA240k-AM. The total amount of polymer in PAA240k-AM hydrogels is 6 g. The PAA3k-AM was synthesized by the same procedure despite of the PAA selection. The P(AA-co-AM) was synthesized with AA, and the total amount of AA and AM is 6 g.

## Cryogenic transmission electron microscopy (Cryo-TEM)

The 25 *wt.%* PAA solution was diluted three times or 5 times with deionized water and then applied on a plasma-cleaned holey carbon grid (R1.2/1.3, 200 mesh; Quantifoil, Cu). The grid was blotted with Vitrobot Mark IV (Thermo Fisher Scientific) using a blot force of −1 and 3 -s blot time at 100% humidity and 9 °C and then plunged into liquid ethane cooled by liquid nitrogen. The grid was inspected on the Cryo-TEM (Tecnai G2 F20 200 kV).

The calculation of chain length is based on the multiply between the length of AA monomer and repeating unit based on molecular weight of PAA. The length of monomer is about 2.17 Å, considering the steric effect of carbon backbone, and repeating units of PAA240k and PAA3k are about 3571 and 41 units, respectively.

## Molecular dynamics simulation (PAA colloids in water)

A periodic model of PAA solution containing two high-molecular-weight PAA (256 repeating units) and water molecules was simulated in the Amorphous Cell module of Materials Studio ver. 2020. The amount of PAA240k chains in water was kept at 20 wt.%. The structure optimization and molecular dynamics were conducted in the Forcite module. A periodic model of PAA solution with 17 low-molecular-weight PAA (30 repeating units) and water molecules was also simulated.

## Attenuated total reflection-Fourier transform infrared spectroscopy (ATR-FTIR)

The FTIR spectrum was conducted by Bruker FTIR spectrometer Matrix-MF from the wavelength 400−4000 nm. The in-situ FTIR spectroscopy was monitored by probing the precursor solution heated in water bath.

## Field-emission scanning electron microscopy (FESEM)

Hydrogels were cutoff with a scissor into small pieces, rapidly plunge froze into liquid ethane to preserve the polymer structure and then freeze-dried to remove vitrified ice. The immersion in liquid ethane and follow-up freeze-drying retain the hydrogel structure. The freeze-dried gels were subjected to gold deposition by thermal evaporator and inspected under FESEM (JSM-7800F Prime, JEOL).

## Atomic force microscopy (AFM)

Hydrogels were prepared with the same procedure as freeze-dried FESEM samples. Freeze-dried samples were examined by atomic force microscope (Omegascope SL, AIST-NT) and approached by the tip (HQ:NSC14/Cr-Au, 5 N/M, Mikromasch). The scanning frequency is 2.0 Hz.

## Small-angle X-ray scattering (SAXS)

The SAXS measurements were conducted at the SSRF beamline BL19U2 (Shanghai, China) at the X-ray energy of 12.0 keV with the wavelength of 1.03 Å. The sample-to-detector distance was 2804.0 mm. For the PAA solutions before and after adding AM, the exposure time was 3 s. The angular conversion between scattering vector q and size d is $q = \frac{2\pi}{d}$. The in-situ SAXS measurements were conducted with a TST 350 tensile device (Linkam Scientific Instruments, UK) fixed on the beamline. Each sample was cyclically loaded at stretch rate of 4 min⁻¹, and SAXS patterns were continuously recorded with the exposure time of 0.9 s and the acquisition time of 1 s. All 2D SAXS patterns were radially averaged into 1D intensity profiles.

Lorentzian peak fitting model is

$$I(q) = \frac{\text{scale}}{\left(1 + \left(\frac{q - q_0}{B}\right)^2\right)} + \text{background} \tag{1}$$

where $q_0$ is the peak position and B is the full width at half-maximum, which is the reciprocal of correlation length.

Correlation length fitting model is[39]

$$I(q) = \frac{A}{q^n} + \frac{C}{1 + (q\xi)^m} + \text{background} \tag{2}$$

where A is the porod scale factor, n is the Porod exponent, C is the Lorentzian scale, ξ is the correlation length and m is Lorentzian exponent. The first term depicts the Porod scattering from clusters and the latter term depicts Lorentzian scattering from polymer chains.

## Rheological measurements

The rheological behavior of hydrogel was examined by rheometer (Anton-Paar MCR 301 rheometer). Frequency-dependent linear viscoelastic response data were obtained at room temperature with the fixed strain ($\gamma$ = 1%) over the frequency range of 0.1–100 rad s⁻¹.

## Tensile measurements

Tensile tests of PAA-AM hydrogels were conducted on a universal testing system (Instron 5966). All samples are approximately 6 mm in length, 5 mm in width, and 1 mm in thickness.

## Self-healing evaluation

The observation of self-healing process was conducted under an industrial microscope. The PAA-AM hydrogel was polymerized on a petri-dish, cut by a razor blade and covered by the PET film to prevent dehydration, waiting for observation after certain self-healing time. The mechanical properties of PAA-AM hydrogels after self-healing were examined after cutting off the hydrogel and covering PET film on damaged interface to prevent dehydration. The direction of tensile test was vertical to that of cutting plane.

## Determination of water content in swelled hydrogel

The water content is estimated based on the original mass of PAA240k-AM hydrogel and weight change from full swelling. The original polymer content of PAA240k-AM is 40 *wt%* regardless of the mass of initiator. Assume all monomers are fully polymerized, the weight change after swelling is attributed to water swelled. The water content of PAA240k-AM after swelling is:

$$\text{water content} = \frac{m_i * 0.4}{(m_i * 0.6 + (m_s - m_i))} \times 100\% \tag{3}$$

where $m_i$ is the initial mass of hydrogel and $m_s$ is the mass of swelled hydrogel.

## Molecular dynamics simulation (intermolecular bonding energies)

A periodic model of PAA-PAM polymer network containing two PAA chains (25 repeating units) and two PAM chains (25 repeating units) was simulated in the Amorphous Cell module of Materials Studio ver. 2020. The structure optimization and the calculation of intermolecular bonding energies were conducted in the Forcite module. A periodic model of PAM polymer network with four PAM chains (25 repeating units) was also simulated.

## Data availability

The data generated in this study have been deposited in the figshare database under accession code dx https://doi.org/10.6084/m9. figshare.28423562 or search code PAA240k-AM Original data. All data are available from the corresponding author upon request.

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

## Acknowledgements

The authors are grateful to the financial support from the Science and Technology Commission of Shanghai Municipality (23DZ1200800) and the Research Fund for Science and Technology on Underwater Vehicles Laboratory (2023-SXJQR-SYSJJ03). The authors thank the support from Analytical Instrumentation Center (#SPST-AIC10112914), SPST, ShanghaiTech. The authors also thank the staff from BL19U2 beamline of National Facility of Protein Science at Shanghai Synchrotron Radiation Facility for assistance during data collection, and Dr. Zhihao Chen, Prof. Shengjie Lin for the support of TST 350 tensile device. This work benefited from the use of the SasView application, originally developed under NSF Award DMR - 0520547. SasView also contains code developed with funding from the EU Horizon 2020 program under the SINE2020 project Grant No 654000.

## Author contributions

Z.S. designed the experiments and finished the draft. R.D. conceived the idea, reviewed and edited the manuscript. Z.Q. performed the Cryo-TEM characterization. P.M. performed the NMR experiments and reviewed the manuscript. W.X. participated in the project. Z.Z. reviewed the manuscript. L.W. revised and gave suggestions about the manuscript. H.F. supervised the project and acquired funding.

## Competing interests

The authors declare no competing interests.
