## [Transparent Peer Review file · Nature Communications]

Observation of Topological Hydrogen-bonding Domains in Physical Hydrogel for Excellent Self-Healing and Elasticity

Corresponding Author: Professor Fuqiang Huang

Version 0:

Reviewer comments:

Reviewer #1

(Remarks to the Author)

Thank you for the opportunity to review this manuscript. Zhang et al. developed a hydrogel composed of polyacrylic acid and polyacrylamide, which exhibits promising self-healing properties and elasticity. As a scientist specializing in fundamental research, I am not very familiar with the applied aspects of this manuscript. However, I believe the manuscript's fundamental sections are lacking in both writing and presentation, as well as in the interpretation and analysis of the experimental data. Specifically, the analysis of the SAXS profiles is particularly inappropriate and incorrect. While the gels may possess excellent mechanical properties, the inclusion of improperly analyzed SAXS data should not be part of the published paper. I cannot recommend this manuscript for publication in its current form. My critical comments are not just because of the high standards of Nat. Commun., but I would also oppose its publication in a lower-impact journal. At the very least, the following points must be addressed before the manuscript can be considered for publication.

1. The terms "cooperative/noncooperative bond/matrix" are difficult to understand within the context of the author's systems. These terms should be defined clearly. Additionally, the PAA colloids should be explained in detail from the start. I found the explanations in the final paragraph of the Introduction and the first paragraph of the Results section insufficient. In Figure 1a, PAA3k appears to be molecularly dispersed linear chains, which are not typically referred to as colloids.
2. This system is a type of double network gel, and there have been numerous papers on double-network gels. Previous studies on double-network gels should be highlighted in the Introduction.
3. In most of the experiments, the polymer concentrations are not mentioned in the main text, making it difficult to fully understand the experimental data.
4. In the SAXS measurements, the authors estimated the radius of gyration (R_g) of PAA. However, these values are not correctly estimated. In aqueous solutions of polyelectrolytes, electrostatic repulsion between the polymers significantly influences the scattering data; in the authors' case, this resulted in much-suppressed scattering intensity in the low- q region of the SAXS profiles. Therefore, the obtained R_g values have no physical meaning. In the Guinier plot, the R_g value should be obtained in the low- q region, but this was not possible due to the aforementioned reason. One possible solution is to estimate the correlation length from the reciprocal q value at the maximum point. Additionally, the scale of the vertical axis of Figures 2a and 2c should be shown, and the SAXS profiles should be displayed as a double-logarithmic plot, as the scaling exponents are important for understanding the data.
5. Figure 2e should be represented as a double-logarithmic plot, as the scaling behavior is crucial for interpreting the rheological data.

Reviewer #2

(Remarks to the Author)

In this work, Zhang et al. presents a compelling approach to the development of a self-healing and elastic physical hydrogel using topological strategies. The authors provide a thorough explanation of the mechanism behind the formation of the colloidal structure, which is well-supported by experimental data. The high self-healing efficiency and remarkable elasticity are impressive and suggest significant potential for practical applications.

1. Figure 2d. It is hard to read the AFM images, it is recommended to point out the mentioned "plains and gullies".
2. The words in Figure 2a and Figure 2c are too small. The author should consider enlarging the font or transfer the related information to a table
3. What is the critical to define the loss of initial shape and the loss of integrity?
4. More discussion on the implications of this work for real-world applications is recommended.
5. More detailed characterization of the hydrogel's mechanical properties under varying environmental conditions would strengthen the manuscript.

Reviewer #3

(Remarks to the Author)

The article by Shaoning Zhang et al. reports a new approach to design self-healing and reversible, yet elastic hydrogels. The authors propose a strategy based on topological hydrogen-bonding domains constructed from polyacrylic acid (PAA) colloids, which are combined with polyacrylamide (PAM) to form a hydrogel with significant self-healing properties and elasticity. They use high-molecular-weight PAA (240k g/mol) to create button-knot colloids, which act as a reversible elastic scaffold, while the PAM provides mechanical strength and ensures structural integrity. The superiority of PAA 240k colloids has been directly compared to PAA 3k which lacks colloidal behaviour.

The hydrogel demonstrates 100% self-healing within 5 hours at room temperature, with the material retaining over 85% toughness and elasticity through repeated mechanical loading cycles. After swelling, it achieves 97.5% water content and maintains structural integrity without residual strain during cyclic tests. These results are substantiated through various techniques, including Tyndall effect experiments, cryo-TEM, rheology, small-angle X-ray scattering (SAXS), and molecular dynamics simulations, which collectively confirm the hydrogel's robust self-healing and elastic properties.

While the study presents well-executed experiments and detailed characterization, approaches involving dual cross-linking and hydrogen bonding interactions of polyacrylic acid (PAA) and polyacrylamide (PAM), are already well-represented in existing literature (Soft Matter, 2013,9, 10287-10293) (J. Mater. Chem. A, 2020,8, 3667-3675) and lead to comparable self-healing and mechanical properties. I therefore recommend publication in a more specialized journal with a polymers or materials focus after the following considerations:

1. The article lacks statements on how the work stands apart from earlier approaches in the introduction. This would help the reader to better position the study.
2. Include chemical structures of both PAA and PAM to allow readers to visualize the chemistry and understand the interactions between PAA colloids and PAM. Without the chemical structures, it is challenging to follow the molecular interactions and the basis for the topological design.
3. In Figure 1c, authors calculate a chain length of 780 nm for PAA240k, but this does not align with the 50 nm scale bar presented in the images. Provide further explanation or correction on how the 780 nm length was determined, ensuring it is consistent with the scale bar.
4. Clarification on FTIR Interpretation (Figure 2b): The statement regarding the shift of the $\nu(\text{COOH})$ peak to a lower wavenumber, indicating complexation between PAA and PAM, is unclear based on Figure 2b. The authors are strongly recommended to revisit this statement, ensuring the observed data accurately support the claim of complexation. Clarifying this could enhance the reader's comprehension for the H-bonding interactions.
5. The "down-turn phenomenon" mentioned lacks definition or explanation, making it unclear for readers. Definition of this term and explanation of its significance within the context of the hydrogel's properties and SAXS data would be helpful for the reader to better follow the data.
6. The relevance of FESEM images to the morphological insights is unclear, and how these correlate with AFM and cryo-TEM findings is not addressed. The authors are recommended to enhance the discussion around the FESEM images and specify how they relate to AFM and cryo-TEM to present a correlated view of the hydrogel morphology.
7. Rheology Results in Figure 2e: Although the hydrogels are described as stable across the frequency sweep, the observed significant stiffness increase at higher frequencies is not explained in the manuscript. Additional commentary on the rheology data to clarify the source of increased stiffness at higher frequencies, particularly in terms of molecular interactions or material properties would help the reader.
8. A claim of 97.5% water content is provided. Supporting data and calculations are needed to substantiate this value, as it is central to the hydrogel's high water retention properties.
9. It is challenging to distinguish which stress-stretch curves were used to calculate the residual strain% values presented. The authors are recommended to identify which data was used for this calculation.
10. The difference in color between samples in Figure 2a is difficult to discern. The authors are recommended to adjust the color scheme for greater contrast, ensuring each sample is easily distinguishable for readability.
11. There are multiple typographical errors (e.g., in Figure S7, Figure S17ab). Please review and edit for correctness.

Reviewer #4

(Remarks to the Author)

I co-reviewed this manuscript with one of the reviewers who provided the listed reports. This is part of the Nature

Communications initiative to facilitate training in peer review and to provide appropriate recognition for Early Career Researchers who co-review manuscripts.

Version 1:

Reviewer comments:

Reviewer #1

(Remarks to the Author)

The authors have revised the manuscript, resulting in an improvement in the analysis of the SAXS data. However, I tend to agree with Referee 3; a more specialized journal would be better suited given the novelty of the concept.

Reviewer #2

(Remarks to the Author)

[Note from the editor: Reviewer 2 looked over both the responses to their own comments, and those from reviewer 3]

It looks reviewer 3's comments have been addressed!

Authors' Response: Rebuttal Letter

Reviewer #1:

Thank you for the opportunity to review this manuscript. Zhang et al. developed a hydrogel composed of polyacrylic acid and polyacrylamide, which exhibits promising self-healing properties and elasticity. As a scientist specializing in fundamental research, I am not very familiar with the applied aspects of this manuscript. However, I believe the manuscript's fundamental sections are lacking in both writing and presentation, as well as in the interpretation and analysis of the experimental data. Specifically, the analysis of the SAXS profiles is particularly inappropriate and incorrect. While the gels may possess excellent mechanical properties, the inclusion of improperly analyzed SAXS data should not be part of the published paper. I cannot recommend this manuscript for publication in its current form. My critical comments are not just because of the high standards of Nat. Commun., but I would also oppose its publication in a lower-impact journal. At the very least, the following points must be addressed before the manuscript can be considered for publication.

Response: Thanks for the helpful suggestions. The issues of Guinier approximation in SAXS profiles have been evaluated, this approximation is indeed inappropriate in case of charged polyelectrolyte solution. Therefore, we take correlation length to evaluate SAXS profiles and the corresponding relations in the structural evolution of PAA240k and PAA3k. We thank again for the valuable advice, and please find point-to-point responses to the comments.

1. The terms "cooperative/noncooperative bond/matrix" are difficult to understand within the context of the author's systems. These terms should be defined clearly. Additionally, the PAA colloids should be explained in detail from the start. I found the explanations in the final paragraph of the Introduction and the first paragraph of the Results section insufficient. In Figure 1a, PAA3k appears to be molecularly dispersed linear chains, which are not typically referred to as colloids.

Response: Thanks for the valuable advice. The cooperative/noncooperative bondings

refer to two types of hydrogen-bonding (H-bonding) interactions. When two H-bondings point along the same direction, these H-bondings strengthen each other, which is called cooperative bonding (Phys. Chem. Chem. Phys., 2016,18, 19557-19566). The carboxylic groups of polyacrylic acid (PAA) form double H-bonding pointing along each other, which indicates the nature of cooperative H-bonding (Figure R1). Meanwhile, the H-bonding between amide groups can point towards either side of N–H bond and thereupon is noncooperative. Additionally, due to the Lewis-acid-base couple between carboxylic group and amide group, the resultant double H-bonding point along each other and strengthen the interaction energy as cooperative H-bonding.

Figure R1. Schematic of H-bonding network consists of the cooperative H-bonding between carboxylic groups and amide groups in hydrogel and the noncooperative H-bonding between amide groups.

Noncooperative matrix represents the region composed of polyacrylamide (PAM), which is dominated by noncooperative H-bonding, and cooperative domains represent the region dominated by cooperative H-bonding.

Statements about PAA colloids are addressed in detail at the beginning of the third paragraph of introduction and the result section.

We apologize for the misinterpretation of schematic for PAA3k. According to the Tyndall effect and Cryo-TEM image, PAA3k solution confirms the colloidal behavior and exhibits the morphology of globular colloids. The corresponding schematic illustration is corrected.

Revision: We selectively utilize the topology of polyacrylic acid (PAA) colloids to assemble double-network hydrogel and locally construct the reversible elastic

cooperative H-bonding domain embedded in noncooperative H-bonding matrix. PAA colloids are assembled via intermolecular cooperative H-bonding in concentrated solution, and different molecular weights lead to distinctive entangled conformation at the same concentration, as elucidated via Tyndall effect, cryo-TEM, small-angle X-ray scattering (SAXS), rheology and simulation. Cooperative H-bonding domain is composed of PAA colloids and polymerized acrylamide (AM) around PAA, contributing to the reversibility of polymer network with strong H-bonding. The excess AM polymerizes aside as noncooperative matrix connecting to the cooperative domain, facilitating the rapidly recoverable polymer network with weak H-bonding. The topology of cooperative domains critically determines the macroscopic properties of hydrogels, and only interconnecting cooperative domains assembled by high-molecular-weight PAA enable hydrogel with a superior self-healing capability (~100% healing in 5 hours at room temperature) and excellent elasticity (< 6.3% residual strain, >100% stress retention, >85% reversibility) upon successive loading.

(Highlighted sentence in the fourth paragraph of Introduction)

The hydrogel was synthesized via the free-radical polymerization of AM along the PAA colloid. PAA is a well-known water-soluble polymer with alternating carbonyl and carboxyl groups and leads to high negative charge density when the carboxyl groups dissociate. The strong negative charges on the PAA backbone induces the repulsive electrostatic screening in aqueous solution and thereupon self-assembles into colloids. Due to the intermolecular cooperative H-bonding and steric alignment, PAA of diverse chain length forms colloid with different structure (Figure 1A). For example, PAA chains with molecular weights of 3,000 (PAA3k) and 240,000 (PAA240k) have chain lengths of approximately 10 nm and 780 nm, respectively, based on the length of monomer and molecular weight. Due to the negligible steric hinderance, low-m.w. PAA substantially contacts with water molecules and thereupon spreads as discrete islands. Meanwhile, high-molecular-weight PAA interacts with each other due to the significant steric hinderance, self-assembling into nanostrings. Upon critical concentration, PAA nanostrings overlap and assemble highly entangled colloids with button-knot morphology (Figure S1), forming button-knot cooperative domains. (Highlighted

sentence in the first paragraph of Result and Discussion)

The cooperative H-bonding consists of double H-bonding between carboxylic groups and amide groups (Figure S2), contributing to high bonding strength. (Highlighted sentence in the second paragraph of Result and Discussion)

The noncooperative H-bonding matrix is instead composed of noncooperative H-bonding between amide groups and thereupon easily dissociates compared to cooperative H-bonding, making cooperative domains the scaffold upon deformation. Due to the discrete distribution, the cooperative domains built by PAA3k chains only reach surrounding PAM chains after polymerization, inducing permanent cut-off damage and continuous disentanglement of the polymer network. Instead, the button-knot cooperative domains by PAA240k colloids become reversible elastic scaffold and stabilize the entire matrix, resulting in highly recoverable mechanical dissipation. Such cooperative domains also knit the mechanical damage at interface, providing the hydrogel with superior self-healing capability. (Highlighted sentence in the second paragraph of Result and Discussion)

The graph depicting the schematic of cooperative H-bonding and noncooperative H-bonding is added as Figure S2.

Figure S2. Schematic of H-bonding network consists of the cooperative H-bonding between carboxylic groups and amide groups in hydrogel and the noncooperative H-bonding between amide groups.

The graph depicting the schematic polymer network of PAA3k-AM is updated as Figure 1A.

Figure 1A. Schematic of the synthesis and stretching of topological hydrogels.

2. This system is a type of double network gel, and there have been numerous papers on double-network gels. Previous studies on double-network gels should be highlighted in the Introduction.

Response: Thanks for the helpful suggestion. Literature about double network gel has already been included in the introduction, and some double network hydrogels have achieved either self-healing capability or elasticity yet cannot surpass the trade-off. Additional introduction about double-network gel is added in the first paragraph of introduction.

Revision: Double-network hydrogels introduce additional polymer network, designed to retain self-assembly from deterioration, yet the self-healing capability and elasticity are still far from satisfactory. (Highlighted sentence in the first paragraph of Introduction)

3. In most of the experiments, the polymer concentrations are not mentioned in the main text, making it difficult to fully understand the experimental data.

Response: Thanks for the careful reading. The polymer concentration is supplemented in the main text.

Revision: 25 wt% PAA240k solution exhibits typical shear-thinning effect at a shear rate of 30 s^{-1} (Figure S2) whereas 25 wt% PAA3k solution is nearly Newtonian liquid, indicating that PAA240k forms colloidal suspension in water. Such PAA240k solution exhibits a more intense Tyndall effect, and PAA240k droplet is highly viscous compared to PAA3k droplet (Figure 1B), indicating the presence of highly entangled

PAA240k colloids. (Highlighted sentence in the third paragraph of Result and Discussion)

Cryo-TEM images of 25 wt% PAA3k and PAA240k solution diluted five times. The actual concentration after blotting is much higher than the initial concentration.

(Highlighted sentence in the caption of Figure 1C)

From SAXS profiles, both PAA3k and PAA240k aqueous solution at 25 wt% experience a downturn at low- q region (Figure 2A), which typically exists in polyanionic polymer solution where the polyanionic backbone carries negative charges and leads electrostatic screening. (Highlighted sentence in the sixth paragraph of Result and Discussion)

4. In the SAXS measurements, the authors estimated the radius of gyration (R_g) of PAA. However, these values are not correctly estimated. In aqueous solutions of polyelectrolytes, electrostatic repulsion between the polymers significantly influences the scattering data; in the authors' case, this resulted in much-suppressed scattering intensity in the low- q region of the SAXS profiles. Therefore, the obtained R_g values have no physical meaning. In the Guinier plot, the R_g value should be obtained in the low- q region, but this was not possible due to the aforementioned reason. One possible solution is to estimate the correlation length from the reciprocal q value at the maximum point. Additionally, the scale of the vertical axis of Figures 2a and 2c should be shown, and the SAXS profiles should be displayed as a double-logarithmic plot, as the scaling exponents are important for understanding the data.

Response: Thanks for the valuable suggestion. The issue about the estimation of the radius of gyration is carefully evaluated, and the Guinier fitting is indeed inappropriate for charged polyelectrolyte solution, which has suppressed signal at low- q region, because Guinier theorem applies to linear relationship between scattering intensity and the square of scattering vector. In this work, PAA240k (polyacrylic acid with molecular weight of 240,000) and PAA3k (polyacrylic acid with molecular weight of 3,000) are charged polyelectrolyte and thereupon require other perspectives to approach the structural information. The review #1 suggests the correlation length from the

reciprocal q value at the characteristic peak, which is evaluated as follows.

The correlation length (ξ) of polyelectrolyte describes the spatial scale over which the electrostatic interaction between counterions and polymer chains are related, and Lorentzian peak fitting allows the estimation of both correlation length and peak position. The Lorentzian peak fitting is based on the formula:

$$I(q) = \frac{scale}{(1+(\frac{q-q_0}{B})^2)} + background \quad (1)$$

where q_0 is the peak position and B is the full width at half-maximum, which is the reciprocal of correlation length. ξ is affected not only by the intermolecular interaction but also by the steric alignment of components in solution.

We applied Lorentzian peak fitting to all SAXS spectra with suppressed scattering intensity at low- q regions (PAA240k solution, PAA3k solution, PAA240k+AM solution, PAA3k+AM solution, PAA240k-AM hydrogel, swelled PAA240k-AM hydrogel and corresponding time-resolved spectra of PAA240k-AM hydrogel and swelled hydrogel). Before polymerization, the ξ of PAA240k precursor solution is 30.3 Å (Figure R3A and Table R1), higher than that of PAA3k precursor ($\xi = 18.9$ Å), because the correlation length scales with the chain length in good solvent and is related to the mesh size of the polymer network in concentrated polymer solutions. Since the content of PAA solution is 25 wt% (~27 g/ml) and induces shear-shinning behavior, PAA solution in this work is considered as concentrated polymer solution. The peak position of PAA240k around 0.042 Å⁻¹ is significantly lower than the calculated chain length of PAA240k (~780 nm) so that this peak is only related to the structure within colloid. The ξ of PAA240k is close to the diameter of PAA240k nanostrings from Cryo-TEM images, in concordance with the observed mesh size of PAA240k assembly. After the addition of AM, the resultant PAA240k+AM precursor retains the characteristic peak around 0.042 Å⁻¹ and leads to higher ξ of 32.3 Å, suggesting enhanced intermolecular interaction and extended state of PAA240k chains upon interacting with AM. Meanwhile, PAA3k solution has the fitting peak at 0.064 Å⁻¹, which is close to the calculated chain length of PAA3k, and thereupon the peak is related to inter-colloid interactions. PAA3k+AM solution results in a characteristic peak around 0.018 Å⁻¹ and is fit to have a ξ of 17.2

Å. The decrease in correlation length suggests the further coiled state of PAA3k chains, which may result from the steric separation by AM monomers and accompanying charge compensation. Therefore, even with strong intermolecular interaction between PAA and AM, the variation of molecular weight significantly affects the assembly state of PAA colloids in mixed solution.

Figure R3. (A) SAXS profiles of PAA240k, PAA240k+AM, PAA3k and PAA3k+AM precursor solution with Lorentzian peak approximation. (B) SAXS profiles of PAA-AM hydrogels with correlation length approximation and Lorentzian peak approximation.

Table R1. The Lorentzian peak fitting parameters of SAXS profiles of PAA240k solution, PAA240k+AM solution, PAA240k-AM hydrogel, PAA3k-AM and PAA3k-AM hydrogel.

Lorentzian peak fit	PAA240k solution	PAA240k +AM precursor	PAA240k -AM hydrogel	PAA3k solution	PAA3k+AM precursor
Scale	378.6±7.2	274.6±6.0	61.6±1.3	28.1±1.6	277.8±4.9
Background	601.6±7.3	515.2±6.2	80.6±1.4	66.3±1.6	29.6±1.2
Peak position (1/Å)	0.0414±0.0001	0.0420±0.0001	0.0435±0.0001	0.0642±0.0002	0.0182±0.0010
Peak HWHM (1/Å)	0.033±0.001	0.031±0.001	0.025±0.001	0.058±0.003	0.053±0.001
Fitting error	3.0	1.9	2.56	1.31	1.34

After the polymerization of AM, PAA240k-AM and PAA3k-AM become double-network hydrogels and exhibit distinctive SAXS profiles. Yet PAA240k-AM hydrogel still retains the characteristic polyelectrolyte peak around 0.04 \AA^{-1} (Figure R3b) while PAA3k-AM hydrogel is dominated by attractive interaction due to further charge

compensation. The ξ of PAA240k-AM is further intensified from 32.3 Å to 40.0 Å after the polymerization, indicating the stronger intermolecular interaction within cooperative domain to sprawl in noncooperative matrix. In the case of PAA3k-AM, correlation length fitting model is instead applied since PAA3k-AM no longer exhibits peak maximum. The fitting function of correlation length model is:

$$I(q) = \frac{A}{q^n} + \frac{C}{1+(q\xi)^m} + background \quad (2)$$

where the first term depicts the Porod scattering from clusters and the latter term depicts Lorentzian scattering from polymer chains. This model fits well with the SAXS profile of PAA3k-AM hydrogel (Figure R3B), and the ξ of PAA3k-AM decreased from 17.2 Å to 12.7 Å after the polymerization (Figure R3 and Table R2), indicating that discrete islands further contract and reach limited PAM chains.

Table R2. The correlation length fitting parameters of SAXS profiles of PAA3k-AM hydrogel.

Correlation length fit	PAA3k-AM hydrogel
Background	10.0±0.7
Lorentz scale, C	0.73±0.01
Porod scale, A	2.1E-7±4.6E-8
Correlation length, ξ	12.73±0.04
Porod exponent, n	3.26±0.05
Lorentz exponent, m	2.07±0.03
Fitting error	0.99

In case of swelled PAA240-AM hydrogel, the Lorentzian peak approximation leads to the increase of ξ from 40.0 Å to 73.0 Å (Figure R4 and Table R5). The larger correlation length indicates that the assembled PAA nanostrings expand upon the introduction of water and leads to highly associated yet entangled PAA nanostrings.

Figure R4. SAXS profiles of PAA240k-AM before and after swelling with Lorentzian peak approximation.

For the time-resolved SAXS profiles, the characteristic peak of PAA3k-AM around 0.04 \AA^{-1} gradually deteriorates (Figure R5), reflecting the inter-colloid ordering continuously decreases. Notably, the ξ of PAA3k-AM remains unchanged throughout cyclic loading (Figure R6 and Table R3), suggesting that PAA3k remains coiled state and is weakly correlated to the deformation. The ξ of PAA240k-AM decreases from 40.0 \AA to 30.3 \AA and stabilizes after the second cycle (Figure R6 and Table R4), conforming with the trend of toughness. After swelling, the characteristic peak of PAA240k-AM barely shifts throughout the cyclic loads, and the ξ of swelled PAA240k-AM increases to 77 \AA and stabilizes (Table R5), indicating that the stability of polymer network is further improved when the chain friction is minimized.

Figure R5. Time-resolved SAXS profiles of (A) PAA3k-AM, (B) PAA240k-AM and (C) swelled PAA240k-AM for ten successive cycles.

Figure R6. The correlation length fit from time-resolved SAXS profiles of PAA3k-AM, PAA240k-AM and swelled PAA240k-AM.

Table R3. The correlation length fitting parameters of SAXS profiles of PAA3k-AM hydrogel cycling at $\lambda = 2$.

Correlation length fit	Virginal	1 st cycle	2 nd cycle	6 th cycle	10 th cycle
Background	10.0±0.7	8.2±0.8	11.2±0.7	9.6±0.8	7.7±0.9
Lorentz scale	0.73±0.01	107.75±1.41	104.70±1.32	103.61±1.50	95.78±1.62
Porod scale	2.1E-7±4.6E-8	6.0E-5±9.5E-6	1.0E-4±1.5E-5	1.E-4±1.4E-5	8.3E-5±1.1E-5
Cor length	12.73±0.04	12.47±0.05	12.56±0.04	12.47±0.05	12.29±0.06
Porod expo	3.26±0.05	3.21±0.03	3.10±0.03	3.12±0.03	3.19±0.03
Lorentz expo	2.07±0.03	2.02±0.03	2.11±0.03	2.00±0.03	1.91±0.03
Fitting error	0.99	0.97	0.98	1.03	1.03

Table R4. The Lorentzian peak fitting parameters of SAXS profiles of PAA240k-AM hydrogel cycling at $\lambda = 2$. The data of initial hydrogel is the same as that in table R1.

Lorentzian peak fit	Virginal	1 st cycle	2 nd cycle	6 th cycle	10 th cycle
Scale	61.6±1.3	138.4±2.6	167.5±2.9	198.7±3.4	168.3±3.1
Background	80.6±1.4	64.4±2.5	58.6±2.6	53.7±2.9	55.8±2.8

Peak position (1/Å)	0.0435±0.0001	0.0340±0.000 3	0.0320±0.0003	0.0294±0.0004	0.0315±0.0003
Peak HWHM (1/Å)	0.025±0.001	0.033±0.001	0.035±0.001	0.037±0.001	0.036±0.001
Fitting error	2.56	1.29	1.53	1.07	1.13

Table R5. The Lorentzian peak fitting parameters of SAXS profiles of swelled PAA240k-AM hydrogel cycling at $\lambda = 2$.

Lorentzian peak fit	Virginal	1 st cycle	2 nd cycle	6 th cycle	10 th cycle
Scale	380.0±0.5	454.1±0.5	561.9±0.5	1071.6±1.3	1115.8±3.0
Background	-0.7±0.2	-2.8±0.2	-5.9±0.2	-41.2±0.7	-60.1±1.5
Peak position (1/Å)	0.0165±0.0001	0.0169±0.0001	0.0167±0.0001	0.0166±0.0001	0.0168±0.0001
Peak HWHM (1/Å)	0.0137±0.0001	0.0131±0.0001	0.0129±0.0001	0.0128±0.0001	0.0132±0.0001
Fitting error	2.4	2.5	3.4	3.0	1.4

In addition, all SAXS profiles are rescaled with double-logarithmic plot (Figure R3, R4 and R5), and data density is reduced to make fit lines more obvious.

Revision: From SAXS profiles, both PAA3k and PAA240k aqueous solution at 25 wt% experience a downturn at low-q region (Figure 2A), which typically exists in polyanionic polymer solution where the polyanionic backbone carries negative charges and leads electrostatic screening. When the separation between polymers is lower than electrostatic screening distance, polyanionic chains self-organize and form loosely ordered structure to minimize such interaction, forming colloids and resulting in a peak at low q region around $q_0 = 0.04 \text{ \AA}^{-1}$ in case of PAA240k solution. (Highlighted sentence in the sixth paragraph of Result and Discussion)

The colloidal structure was further evaluated by Lorentzian peak fitting to obtain the correlation length (ξ), which describes the spatial extent over which the electrostatic interactions between the charged groups on the polymer chain are significant. The ξ of PAA240k precursor solution is 30.3 Å (Figure 2A and Table S1), higher than that of PAA3k precursor ($\xi = 18.9 \text{ \AA}$), because the correlation length scales with the chain length in good solvent and is related to the mesh size of the polymer network in

concentrated polymer solutions. Additionally, the peak position of PAA240k is significantly lower than calculated chain length of PAA240k, indicating that the peak is attributed to the structure within colloid. The ξ of PAA240k is close to the diameter of PAA240k nanostrings from Cryo-TEM images, in concordance with the observed mesh size of PAA240k assembly. After the addition of AM, the resultant PAA240k+AM precursor retains the characteristic peak around 0.042 \AA^{-1} and leads to higher ξ of 32.3 \AA , suggesting enhanced intermolecular interaction and extended state of PAA240k chains upon interacting with AM. Meanwhile, PAA3k solution has the fitting peak at 0.064 \AA^{-1} (Table S1), which is close to the calculated chain length of PAA3k, and thereupon the peak is related to inter-colloid interactions. PAA3k+AM solution results in a characteristic peak around 0.018 \AA^{-1} and is fit to have a ξ of 17.2 \AA . The decrease in correlation length suggests the further coiled state of PAA3k chains, which may result from the steric separation by AM monomers and accompanying charge compensation.

(Highlighted sentence in the seventh paragraph of Result and Discussion)

The ξ of PAA240k-AM is further intensified from 32.3 \AA to 40.0 \AA after the polymerization (Figure 2A, 2C and Table S1), indicating the stronger intermolecular interaction within cooperative domain to sprawl in noncooperative matrix. Meanwhile, the ξ of PAA3k-AM decreased from 17.2 \AA to 12.7 \AA after the polymerization (Table S3), indicating that discrete islands further contract and reach limited PAM chains.

(Highlighted sentence in the ninth paragraph of Result and Discussion)

The Lorentzian peak approximation leads to the increase of ξ from 40.0 \AA to 73.0 \AA (Table S5), indicating that the assembled PAA nanostrings expand upon the introduction of water and leads to highly associated yet entangled PAA nanostrings. (Highlighted sentence in the fifteenth paragraph of Result and Discussion)

The ξ of PAA240k-AM decreases from 40.0 \AA to 30.3 \AA and stabilizes after the second cycle (Figure S23 and Table S4), conforming with the trend of toughness. Meanwhile, the ξ of PAA3k-AM continuously decreases after the second cycle, reflecting that cooperative domains within PAA3k-AM substantially contact. (Highlighted sentence in the eighteenth paragraph of Result and Discussion)

After swelling, the characteristic peak of PAA240k-AM barely shifts throughout the cyclic loads, and the ξ of swelled PAA240k-AM increases to 77 Å and stabilizes (Figure 5E, S23 and Table S5), indicating that the stability of polymer network is further improved when the chain friction is minimized. (Highlighted sentence in the eighteenth paragraph of Result and Discussion)

Lorentzian peak fitting model is:

$$I(q) = \frac{scale}{(1+(\frac{q-q_0}{B})^2)} + background$$

where q_0 is the peak position and B is the full width at half-maximum, which is the reciprocal of correlation length.

Correlation length fitting model is:

$$I(q) = \frac{A}{q^n} + \frac{C}{1 + (q\xi)^m} + background$$

where A is the porod scale factor, n is the Porod exponent, C is the Lorentzian scale, ξ is the correlation length and m is Lorentzian exponent. The first term depicts the Porod scattering from clusters and the latter term depicts Lorentzian scattering from polymer chains. (Highlighted sentence in the SAXS paragraph of experimental procedures)

The graph depicting SAXS profiles of PAA240k, PAA240+AM, PAA3k and PAA3k+AM precursor with Lorentzian approximation is updated as Figure 2A.

Figure 2. (A) SAXS profiles of PAA240k, PAA240k+AM, PAA3k and PAA3k+AM precursor solution with Lorentzian peak approximation.

The graph depicting SAXS profiles of PAA240k-AM and PAA3k-AM hydrogels with Lorentzian approximation is updated as Figure 2C.

Figure 2. (C) SAXS profiles of PAA240k-AM hydrogels with Lorentzian peak approximation.

The graph depicting SAXS profiles of PAA240k-AM and PAA3k-AM hydrogels with Lorentzian approximation is updated as Figure 4C.

Figure 4. (C) SAXS profiles of PAA240k-AM before and after swelling with Lorentzian peak approximation.

The graph depicting time-resolved SAXS profiles of (C) PAA3k-AM, (D) PAA240k-AM and (E) swelled PAA240k-AM for ten successive cycles. is updated in Figure 5.

Figure 5. (C-E) Time-resolved SAXS profiles of (C) PAA3k-AM, (D) PAA240k-AM and (E) swelled PAA240k-AM for ten successive cycles.

The graph depicting the evolution of correlation length of PAA240k-AM and swelled PAA240k-AM for ten successive cycles is updated as Figure S22.

Figure S23. The correlation length fit from time-resolved SAXS profiles of PAA3k-AM, PAA240k-AM and swelled PAA240k-AM.

The table listing Lorentzian peaking fitting parameters of PAA240k solution, PAA240k+AM solution, PAA240k-AM hydrogel, PAA3k-AM and PAA3k-AM hydrogel has been updated in Table S1.

Table S1. The Lorentzian peak fitting parameters of SAXS profiles of PAA240k solution, PAA240k+AM solution, PAA240k-AM hydrogel, PAA3k-AM and PAA3k-AM hydrogel.

Lorentzian peak fit	PAA240k solution	PAA240k +AM precursor	PAA240k -AM hydrogel	PAA3k solution	PAA3k-AM precursor
Scale	378.6±7.2	274.6±6.0	61.6±1.3	28.1±1.6	277.8±4.9
Background	601.6±7.3	515.2±6.2	80.6±1.4	66.3±1.6	29.6±1.2
Peak position (1/Å)	0.0414±0.0001	0.0420±0.0001	0.0435±0.0001	0.0642±0.0002	0.0182±0.0010
Peak HWHM (1/Å)	0.033±0.001	0.031±0.001	0.025±0.001	0.058±0.003	0.053±0.001
Fitting error	3.0	1.9	2.56	1.31	1.34

The table listing correlation length fitting parameters of PAA3k-AM hydrogel cycling at $\lambda = 2$ has been updated in Table S3.

Table S3 The correlation length fitting parameters of SAXS profiles of PAA3k-AM hydrogel cycling at $\lambda = 2$.

Correlation length fit	Virginal	1 st cycle	2 nd cycle	6 th cycle	10 th cycle
----------	-----------------------	-----------------------	-----------------------	------------------------

Background	10.0±0.7	8.2±0.8	11.2±0.7	9.6±0.8	7.7±0.9
Lorentz scale, C	0.73±0.01	107.75±1.41	104.70±1.32	103.61±1.50	95.78±1.62
Porod scale, A	2.1E-7±4.6E-8	6.0E-5±9.5E-6	1.0E-4±1.5E-5	1.E-4±1.4E-5	8.3E-5±1.1E-5
Correlation length, ξ	12.73±0.04	12.47±0.05	12.56±0.04	12.47±0.05	12.29±0.06
Porod exponent, n	3.26±0.05	3.21±0.03	3.10±0.03	3.12±0.03	3.19±0.03
Lorentz exponent, m	2.07±0.03	2.02±0.03	2.11±0.03	2.00±0.03	1.91±0.03
Fitting error	0.99	0.97	0.98	1.03	1.03

The table listing Lorentzian peak fitting parameters of PAA240k-AM hydrogel cycling at $\lambda = 2$ has been updated in Table S4.

Table S4 The Lorentzian peak fitting parameters of SAXS profiles of PAA240k-AM hydrogel cycling at $\lambda = 2$. The data of initial hydrogel is the same as that in table S1.

Lorentzian peak fit	Virginal	1 st cycle	2 nd cycle	6 th cycle	10 th cycle
Scale	61.6±1.3	138.4±2.6	167.5±2.9	198.7±3.4	168.3±3.1
Background	80.6±1.4	64.4±2.5	58.6±2.6	53.7±2.9	55.8±2.8
Peak position (1/Å)	0.0435±0.0001	0.0340±0.0003	0.0320±0.0003	0.0294±0.0004	0.0315±0.0003
Peak HWHM (1/Å)	0.025±0.001	0.033±0.001	0.035±0.001	0.037±0.001	0.036±0.001
Fitting error	2.56	1.29	1.53	1.07	1.13

The table listing Lorentzian peak fitting parameters of swelled PAA240k-AM hydrogel cycling at $\lambda = 2$ has been updated in Table S5.

Table S5. The Lorentzian peak fitting parameters of SAXS profiles of swelled PAA240k-AM hydrogel cycling at $\lambda = 2$.

Lorentzian peak fit	Virginal	1 st cycle	2 nd cycle	6 th cycle	10 th cycle
Scale	380.0±0.5	454.1±0.5	561.9±0.5	1071.6±1.3	1115.8±3.0
Background	-0.7±0.2	-2.8±0.2	-5.9±0.2	-41.2±0.7	-60.1±1.5
Peak position (1/Å)	0.0165±0.0001	0.0169±0.0001	0.0167±0.0001	0.0166±0.0001	0.0168±0.0001

Peak HWHM (1/Å)	0.0137±0.0001	0.0131±0.0001	0.0129±0.0001	0.0128±0.0001	0.0132±0.0001
Fitting error	2.4	2.5	3.4	3.0	1.4

Reviewer #2:

In this work, Zhang et al. presents a compelling approach to the development of a self-healing and elastic physical hydrogel using topological strategies. The authors provide a thorough explanation of the mechanism behind the formation of the colloidal structure, which is well-supported by experimental data. The high self-healing efficiency and remarkable elasticity are impressive and suggest significant potential for practical applications.

Response: We thank the kind comments from Reviewer #2.

1. Figure 2d. It is hard to read the AFM images, it is recommended to point out the mentioned plains and gullies.

Response: Thanks for the helpful suggestion. Plains and Gullies were labeled in Figure 2D.

Revision: Plains and Gullies of PAA3k-AM and PAA240k-AM were drawn with white dashed lines and labeled with corresponding structures in Figure 2D.

Figure 2D. AFM images of PAA3k-AM and PAA240k-AM.

2. The words in Figure 2a and Figure 2c are too small. The author should consider enlarging the font or transfer the related information to a table

Response: Thanks for the advise. The fonts of Figure 2a and Figure 2c were enlarged, and corresponding fitting information is listed in Table S1 and S3.

Revision: All fonts in Figure 2a and 2c were enlarged.

Figure 2A. SAXS profiles of PAA240k, PAA240k+AM, PAA3k and PAA3k+AM precursor solution with Lorentzian peak approximation.

Figure 2C. SAXS profiles of PAA240k-AM hydrogels with Lorentzian peak approximation.

3. What is the critical to define the loss of initial shape and the loss of integrity?

Response: Thanks for the valuable question. The initial shapes of P(AA-co-AM), PAA3k-AM and PAA240k-AM are all rectangular, and the loss of initial shape takes place when the shape of swelling hydrogel is no longer rectangular. PAA240k-AM always retains rectangular shape throughout the swelling process while the edge of swelling PAA3k-AM becomes irregular, which may result from the partial dissolution of hydrogel into water. In case of the loss of integrity, both P(AA-co-AM) and PAA3k-AM are unable to separate from water tank, which probably results from the high dissociation of PAA3k chains and the followed de-entanglement of hydrogel network.

Revision: None.

4. More discussion on the implications of this work for real-world applications is recommended.

Response: Thanks for the suggestion. More discussion about the importance of molecular weight and the implication for real-world applications is added in the manuscript.

Revision: Based on the structural evolution of two PAA-AM hydrogels, molecular weight significantly impacts the packing conformation of PAA in multi-component concentrated solution and gel. Low-m.w. PAA exhibits coiled state and is separated by PAM network, thereby losing the interaction among PAA colloids. High-molecular-weight PAA instead are assembled as flexible nanostrings and form interconnecting scaffold, securing the self-healing network and structural stability. Therefore, the self-assembly conformation of PAA determines the reconstruction of polymer network and the structural integrity during deformation. In real-world applications, mechano-stability is critical for the robustness of devices, and autonomously self-healing capability prevents devices from unexpected damage. The hydrogel in this work provides strategy to surpass the trade-off between self-healing capability and elasticity, potentially solving the requirement of next-generation applications. (Highlighted sentence in the nineteenth paragraph of Result and discussion)

5. More detailed characterization of the hydrogel's mechanical properties under varying environmental conditions would strengthen the manuscript.

Response: Thanks for the valuable suggestion. Both tensile fracture and cyclic loading of PAA240k-AM hydrogel were conducted under different humidities to supplement the mechanical properties. The previous mechanical properties of PAA-AM hydrogels were conducted at ~60% humidity, and therefore, supplementary mechanical properties of PAA240k-AM hydrogels were examined at 30% and 90% humidity (Figure R7). The breaking stress increases with lower humidity while the maximum stretchability is instead compensated at low humidity. For cyclic loading at 30% humidity, PAA240k-

AM has unstable loading curve even though no residual strain is observed (Figure R8A) while PAA240k-AM exhibit similar cyclic performance in 90% humidity to that in 60% humidity (Figure R8B).

Figure R7. Stress-stretch curves of PAA240k-AM at different humidities. The stretch rate is 2 min^{-1} .

Figure R8. Successive stress-stretch curves of PAA240k-AM at (A) 30% and (B) 90% humidities at $\lambda = 2$ for 10 cycles. The stretch rate is 3 min^{-1} .

Additionally, the self-healing performance of PAA240k-AM was examined at higher temperature. Due to enhanced interdiffusion at higher temperature, PAA240-AM hydrogel achieves full healing capability in one hour.

Figure R9. Stress-stretch curves of PAA240k-AM after healing at 333 K. The stretch rate is 2 min^{-1} .

Revision: PAA240k-AM shows humidity-dependent stretch fracture behavior and possesses the stretchability of 12 and maximum stress of 110 kPa at 60% humidity (Figure S9), while PAA3k-AM and P(AA-co-AM) only demonstrate limited stretchability and breakage stress (Figure S10). (Highlighted sentence in the twelfth paragraph of Result and Discussion)

At 333 K, the self-healing efficiency of PAA240k-AM is further improved, completely restoring mechanical behavior in one hour (Figure S12). (Highlighted sentence in the twelfth paragraph of Result and Discussion)

Successive stretch cycles at different humidities shows that PAA240k-AM is capable of bouncing back, dissipating energy along the loading cycle and achieving the same stress as that of the first cycle above 60% humidity (Figure 3F and S15). (Highlighted sentence in the thirteenth paragraph of Result and Discussion)

The graph depicting the stress-stretch curves of PAA240k-AM at different humidities is updated as Figure S9.

Figure S10. Stress-strain curves of PAA240k-AM at different humidities. The stretch rate is 2 min^{-1} .

The graph depicting the stress-stretch curves of PAA240k-AM after healing at 333K is updated as Figure S12.

Figure S12. Stress-stretch curves of PAA240k-AM after different healing time at 333K. The stretch rate is 2 min^{-1} .

The graph depicting the successive stress-stretch curves of PAA240k-AM at $\lambda = 2$ for 10 cycles is updated as Figure S15.

Figure S15. Successive stress-stretch curves of PAA240k-AM at (A) 30% humidity and (B) 90% humidity at $\lambda = 2$ for 10 cycles. The stretch rate is 3 min^{-1} .

Reviewer #3:

The article by Shaoning Zhang et al. reports a new approach to design self-healing and reversible, yet elastic hydrogels. The authors propose a strategy based on topological hydrogen-bonding domains constructed from polyacrylic acid (PAA) colloids, which are combined with polyacrylamide (PAM) to form a hydrogel with significant self-healing properties and elasticity. They use high-molecular-weight PAA (240k g/mol) to create button-knot colloids, which act as a reversible elastic scaffold, while the PAM provides mechanical strength and ensures structural integrity. The superiority of PAA 240k colloids has been directly compared to PAA 3k which lacks colloidal behaviour. The hydrogel demonstrates 100% self-healing within 5 hours at room temperature, with the material retaining over 85% toughness and elasticity through repeated mechanical loading cycles. After swelling, it achieves 97.5% water content and maintains structural integrity without residual strain during cyclic tests. These results are substantiated through various techniques, including Tyndall effect experiments, cryo-TEM, rheology, small-angle X-ray scattering (SAXS), and molecular dynamics simulations, which collectively confirm the hydrogel's robust self-healing and elastic properties.

While the study presents well-executed experiments and detailed characterization, approaches involving dual cross-linking and hydrogen bonding interactions of

polyacrylic acid (PAA) and polyacrylamide (PAM), are already well-represented in existing literature (Soft Matter, 2013,9, 10287-10293) (J. Mater. Chem. A, 2020,8, 3667-3675) and lead to comparable self-healing and mechanical properties. I therefore recommend publication in a more specialized journal with a polymers or materials focus after the following considerations:

Response: We thank the kindly comments from Reviewer #3. The key concept in this work stems from the topology of polymer network, which is the assembly of polymer chains due to weak interaction and steric hinderance, and the topology significantly decides the self-healing capability and elasticity upon cyclic deformation. PAA colloids lead to interconnecting button-knot scaffold only at high chain length, and subsequent polymerization of AM leads to further entanglement of PAM around PAA colloidal scaffold. The excellent self-healing capability results from not only hydrogen-bonding interaction but also the interconnecting PAA scaffold to restore structural integrity after polymer diffusion.

Elasticity also depends on both the topology of polymer network and hydrogen bonding interaction. We further evaluate the crosslinking degree of PAA-AM hydrogels based on rheological properties (Figure R10), and the slopes of PAA240k-AM and PAA3k-AM are 0.96 and 0.94, which indicates the high degree of physically crosslinking nature in both hydrogels. However, the initial residual strain and further propagation through cyclic loading reflect the different elasticity, again suggesting the significance of topology for macroscopic properties of hydrogels.

Figure R10. Complex viscosity (η^*) as a function of frequency (ω) for PAA240k-AM, PAA3k-AM and P(AA-co-AM) hydrogels.

The hydrogels mentioned by the reviewer rather have different topologies of polymer network compared to this work. Both works introduce the hydrophobic assembly as supporting scaffold and hydrophilic polymer chains as dissipating region, and thereupon strong interactions (cooperative hydrogen bonding and ionic interaction) dissipate between hydrophilic chains, which require time to recover during successive loading. These hydrogels indeed exhibit self-healing capabilities, yet the performance of self-healing behavior and elasticity is still far from satisfactory. The detailed performance is listed in following table.

Table R6. Comparison of self-healing capability, toughness retention and residual strain of PAAc, PAAm/PAA-1.1%Fe³⁺/NaCl and PAA240k-AM hydrogels.

Hydrogel	Self-healing capability	Toughness retention after 10 cycles	Residual strain at 10 th cycle
PAAc (C ₀ = 30%) ^a	20% after 0.5 hour 50% after 0.5 hour (at 333 K)	Data only after resting for 10 mins	
PAAm/PAA-1.1%Fe ³⁺ /NaCl ^b	90% after 24 hours	63%	33%
This work	100% after 5 hours 100% after 1 hour (at 333 K)	85%	6.3%

a. Hydrogel from Soft Matter, 2013,9, 10287-10293.

b. Hydrogel from J. Mater. Chem. A, 2020,8, 3667-3675

Therefore, the self-healing capability and cyclic performance of PAA240k-AM in this work is superior to those of PAAm/PAA-1.1%Fe³⁺/NaCl (J. Mater. Chem. A, 2020,8, 3667-3675). In case of PAAc hydrogel (Soft Matter, 2013,9, 10287-10293), the cyclic performance was performed only after resting for 10 mins and was not comparable based on the literature data. However, extra resting time indicates the slow recovery of polymer network from deformation, reflecting the inferior elasticity of PAAc hydrogel. To further examine the elasticity of PAAc hydrogel, PAAc hydrogel was repeated based on the reported experimental procedure. The tensile fracture curve of repeated PAAc hydrogel almost coincides with that from literature (Figure R11A), proving the

successful repetition. Then successive loading curve was conducted on PAAc hydrogel at $\lambda = 2$ with no resting time (Figure R11B). Significant deterioration of toughness (60.7% after 10 cycles) and residual strain (26.4% at the 10th cycle) is observed, again proving the inferior elasticity of PAAc hydrogel. In addition, the self-healing performance of PAA240k-AM was further examined at 333 K (Figure R12), and complete healing was achieved after 1 hour. Therefore, PAA240k-AM hydrogel in this work has outstanding performance compared to both reported hydrogels.

Figure R11. (A) stress-stretch curve of repeated PAAc hydrogel and digitized PAAc hydrogel in literature ($c_0 = 30$ w/v% AAc). The stretch rate is 8 min^{-1} . (B) Successive stress-stretch curves of repeated PAAc hydrogel at $\lambda = 2$ with no resting time. The stretch rate is 3 min^{-1} .

Figure R12. Stress-stretch curves of PAA240k-AM after healing at 333 K. The stretch rate is 2 min^{-1} .

Revision:

The topology of a polymer network stems from the assembly of polymer chains due to intermolecular interactions and steric hinderance and plays important roles in

macroscopic properties, in which multiple functionalities can be regulated. For instance, hydrophobic assembly by NaCl solution reconstructed the polymer network from the random distribution into certain topology where discrete hydrophobic domains hold ionic polyelectrolyte chains, leading to 90% self-healing capability within 24 hours and recovering the full mechanical property after resting for 4 mins. Yet the high strength of cooperative H-bonding and ionic interaction between polyelectrolyte chains deteriorates the polymer network during successive loading. (Highlighted sentence in the third paragraph of Introduction)

Literature from J. Mater. Chem. A, 2020,8, 3667-3675 has been added in the table of performance comparison.

The graph depicting the stress-stretch curves of PAA240k-AM after healing at 333K is updated as Figure S11.

Figure S12. Stress-stretch curves of PAA240k-AM after different healing time at 333K. The stretch rate is 2 min⁻¹.

1. The article lacks statements on how the work stands apart from earlier approaches in the introduction. This would help the reader to better position the study.

Response: Thanks for the suggestion. The key concept of this work lies in the topology of network, which significantly affects the macroscopic properties of hydrogel. Previous literature primarily focuses on intermolecular interaction and crosslinks within hydrogel while topology further includes the assembly of polymer chains. To elucidate the importance of topology, additional statements about the topology of

polymer network and one example of hydrophobic assembly are first added in the third paragraph of introduction (J. Mater. Chem. A, 2020,8, 3667-3675). Then the fourth paragraph of introduction is re-written to elucidate the importance of colloidal conformation of PAA and the corresponding influence of topology and intermolecular interaction.

Additional statements are added in the FESEM section. PAA240k-AM hydrogel exhibits homogeneous morphology at macroscopical level, while typical hydrogels in literature consist of mesopores and micropores. (Adv. Mater. 2016, 28, 7178–7184, J. Mater. Chem. A, 2019, 7, 24814–24829, Ind. Eng. Chem. Res. 2019, 58, 17001–17009, etc.)

Literature comparison of reversibility and self-healing capability is presented to compare the performance of PAA240k-AM with that of other hydrogels (Figure R13).

Figure R13. Comparison of self-healing efficiency and toughness retention between this work and previously reported hydrogels. Solid symbols belong to non-self-healing hydrogels, and hollow symbols are subtractive self-healing hydrogels.

Revision: The topology of a polymer network stems from the assembly of polymer chains due to intermolecular interactions and steric hinderance and plays important roles in macroscopic properties, in which multiple functionalities can be regulated. For instance, hydrophobic assembly by NaCl solution reconstructed the polymer network from the random distribution into certain topology where discrete hydrophobic domains

hold ionic polyelectrolyte chains, leading to 90% self-healing capability within 24 hours and recovering the full mechanical property after resting for 4 mins. Yet the high strength of cooperative H-bonding and ionic interaction between polyelectrolyte chains deteriorates the polymer network during successive loading. (Highlighted sentence in the third paragraph of Introduction)

We selectively utilize the topology of polyacrylic acid (PAA) colloids to assemble double-network hydrogel and locally construct the reversible elastic cooperative H-bonding domain embedded in noncooperative H-bonding matrix. PAA colloids are assembled via intermolecular cooperative H-bonding in concentrated solution, and different molecular weights lead to distinctive entangled conformation at the same concentration, as elucidated via Tyndall effect, cryo-TEM, small-angle X-ray scattering (SAXS), rheology and simulation. Cooperative H-bonding domain is composed of PAA colloids and polymerized acrylamide (AM) around PAA, contributing to the reversibility of polymer network with strong H-bonding. The excess AM polymerizes aside as noncooperative matrix connecting to the cooperative domain, facilitating the rapidly recoverable polymer network with weak H-bonding. The topology of cooperative domains critically determines the macroscopic properties of hydrogels, and only interconnecting cooperative domains assembled by high-molecular-weight PAA enable hydrogel with a superior self-healing capability (~100% healing in 5 hours at room temperature) and excellent elasticity (< 6.3% residual strain, >100% stress retention, >85% reversibility) upon successive loading. (Highlighted sentence in the fourth paragraph of Introduction)

Field-emission scanning electronic microscope (FESEM) images show that PAA240k-AM has uniform morphology at micron level, which is different from typical hydrogels with mesopores or micropores, while PAA3k-AM leads to bumpy surface (Figure S6). (Highlighted sentence in the tenth paragraph of Result and Discussion)

2. Include chemical structures of both PAA and PAM to allow readers to visualize the chemistry and understand the interactions between PAA colloids and PAM. Without the chemical structures, it is challenging to follow the molecular interactions and the basis

for the topological design.

Response: Thanks for the advice. The schematic of chemical structure and related intermolecular interaction was supplemented in the beginning of supplementary information (Figure R14).

Figure R14. Schematic of H-bonding network consists of the cooperative H-bonding between carboxylic groups and amide groups in hydrogel and the noncooperative H-bonding between amide groups.

Revision: The graph depicting the schematic of cooperative H-bonding and noncooperative H-bonding is added as Figure S2.

Figure S2. Schematic of H-bonding network consists of the cooperative H-bonding between carboxylic groups and amide groups in hydrogel and the noncooperative H-bonding between amide groups.

3. In Figure 1c, authors calculate a chain length of 780 nm for PAA240k, but this does not align with the 50 nm scale bar presented in the images. Provide further explanation or correction on how the 780 nm length was determined, ensuring it is consistent with

the scale bar.

Response: Thanks for the careful reading. The chain length here is calculated based on the length of a monomer, which is about 2.18 Å, and the molecular weight of PAA, and the steric alignment of carbon backbone is considered. The aim of this calculation is to state the significant difference of chain length between PAA240k and PAA3k. Notably, the diameter of one PAA240k is either 0.77 Å or 3.24 Å depending on the steric alignment of functional group. Therefore, the nanostrings in Figure 1C is rather the colloidal assembly of PAA240k chains, which have diameter of ~4 nm. To clarify the potential misunderstanding, the explanation about the calculation and the dimension of one PAA chain is added in manuscript and supplementary information.

Revision: For example, PAA chains with molecular weights of 3,000 (PAA3k) and 240,000 (PAA240k) have chain lengths of approximately 10 nm and 780 nm, respectively, based on the length of monomer (~2.18 Å) and molecular weight.

(Highlighted sentence in the first paragraph of Result and Discussion)

The calculation of chain length is based on the multiply between the length of AA monomer and repeating unit based on molecular weight of PAA. The length of monomer is about 2.17 Å, considering the steric effect of carbon backbone, and repeating units of PAA240k and PAA3k are about 3571 and 41 units, respectively.

(Highlighted sentence in the Cryo-TEM section of Experimental procedures)

4. Clarification on FTIR Interpretation (Figure 2b): The statement regarding the shift of the $\nu(\text{COOH})$ peak to a lower wavenumber, indicating complexation between PAA and PAM, is unclear based on Figure 2b. The authors are strongly recommended to revisit this statement, ensuring the observed data accurately support the claim of complexation. Clarifying this could enhance the reader's comprehension for the H-bonding interactions.

Response: Thanks for the suggestion. The shift in Figure 2b is not obvious because the polymerization is conducted at 333 K. At high temperature, the hydrogen bonding between PAA and AM is partially dissociated so that FTIR signal from interpolymer complex is not obvious. Therefore, we supplement the FTIR spectrum of PAA240k-

AM hydrogel at 298 K (Figure R15). As temperature decreases, the hydrogen bonding repopulates, and the $\nu(\text{COOH})$ shifts towards lower wavenumber, confirming the interpolymer complex between PAA and PAM.

Figure R15. ATR-FTIR spectra of PAA240k-AM hydrogel at 333 K and 298 K.

The corresponding interpretation and figure are added in the manuscript and supplementary information.

Revision: The graph depicting the FTIR spectra of PAA240k-AM hydrogel at different temperature is added as Figure S4.

Figure S4. ATR-FTIR spectra of PAA240k-AM hydrogel at 333 K and 298 K.

5. The "down-turn phenomenon" mentioned lacks definition or explanation, making it unclear for readers. Definition of this term and explanation of its significance within the context of the hydrogel's properties and SAXS data would be helpful for the reader to better follow the data.

Response: Thanks for the helpful suggestion. The 'down-turn phenomenon' is the

indication of polyelectrolyte solution at low ionic strength. Polyelectrolytes are polymers that carry charges, and the resultant electrostatic interaction is screened via counterions in polyelectrolyte solution. When the concentration of solution passes dilute region, i.e. the distance between polyelectrolyte chains is lower than electrostatic screening distance, the chain conformation becomes more ordered assembly. Such homogeneous distribution of polyelectrolyte assembly leads to much suppressed scattering signal at low- q region, which is expressed as ‘downturn’ in the manuscript. More explanation about this phenomenon is added in the manuscript.

Revision: From SAXS profiles, both PAA3k and PAA240k aqueous solution at 25 wt% experience a downturn at low- q region (Figure 2A), which typically exists in polyanionic polymer solution where the polyanionic backbone carries negative charges and leads electrostatic screening. When the separation between polymers is lower than electrostatic screening distance, polyanionic chains self-organize and form loosely ordered structure to minimize such interaction, forming colloids and resulting in a peak at low q region around $q_0 = 0.04 \text{ \AA}^{-1}$ in case of PAA240k solution. (Highlighted sentence in the sixth paragraph of Result and Discussion)

6. The relevance of FESEM images to the morphological insights is unclear, and how these correlate with AFM and cryo-TEM findings is not addressed. The authors are recommended to enhance the discussion around the FESEM images and specify how they relate to AFM and cryo-TEM to present a correlated view of the hydrogel morphology.

Response: Thanks for the advice. The primary objective of FESEM images is to provide the macroscopic morphology of PAA-AM hydrogels, and these hydrogels are homogeneous without mesopore and micropore, which is different from hydrogels from other literature (Adv. Mater. 2016, 28, 7178–7184, J. Mater. Chem. A, 2019, 7, 24814–24829, Ind. Eng. Chem. Res. 2019, 58, 17001–17009, etc.). Both FESEM and AFM images originate from the same PAA-AM hydrogel. FESEM is capable of observing

morphology of large scale yet hard to inspect insulated sample at nanoscale even with platinum sputtering. AFM instead provides nanoscale morphology and depth profile and is complementary with FESEM.

In order to correlate FESEM images with AFM images and Cryo-TEM images, AFM images with larger scale are provided (Figure R16). PAA3k-AM demonstrates uniformity at nanoscale due to the sub-nano globular colloids of PAA3k, which is observed from Cryo-TEM images, yet exhibits variant morphology at microscale, which is consistent with the bumpy surface observed from FESEM image (Figure R17). Meanwhile, PAA240k-AM consistently has interconnecting plains only at nanoscale, conforming with the homogeneous morphology at microscale from FESEM image.

Figure R16. AFM images of (A-B) PAA240k-AM and (C-D) PAA3k-AM.

Figure R17. FESEM images of (A) PAA240k-AM and (B) PAA3k-AM.

Revision: Field-emission scanning electronic microscope (FESEM) images shows that PAA240k-AM has uniform morphology at micron level, which is different from typical hydrogels with mesopores or micropores, while PAA3k-AM leads to bumpy surface (Figure S6). (Highlighted sentence in the tenth paragraph of Result and Discussion)

PAA3k-AM has dense yet tiny plains at nanoscale, and PAA240k-AM leads to interconnecting and stout plains, again indicating that PAA240k-AM has continuous PAA scaffold, but PAA3k-AM has discrete domains. (Highlighted sentence in the tenth paragraph of Result and Discussion)

7. Rheology Results in Figure 2e: Although the hydrogels are described as stable across the frequency sweep, the observed significant stiffness increase at higher frequencies is not explained in the manuscript. Additional commentary on the rheology data to clarify the source of increased stiffness at higher frequencies, particularly in terms of molecular interactions or material properties would help the reader.

Response: Thanks for the advice. The increase of stiffness at higher frequencies indicates the entangled and non-permanent nature of polymer network, which is observed in both polymer melts and gels. In an entangled polymer network, polymer chains are intertwined with each other. At low frequencies, the polymer chains have enough time to relax and rearrange. At higher frequencies, the time scale of the applied deformation is very short so that polymer chains are forced to store the applied energy

elastically, leading to higher elastic modulus. Additional comment on the increase of stiffness is added in the manuscript.

Revision: Specially, the G' of PAA240k-AM reaches 61 kPa at $\omega = 1 \text{ rad s}^{-1}$, even higher than the value of reported chemically crosslinked PAM hydrogel, and is increasing across the frequency sweep. The increase of stiffness at higher frequencies indicates the entangled and non-permanent nature of polymer network, which is observed in both polymer melts and gels. (Highlighted sentence in the eleventh paragraph of Result and Discussion)

8. A claim of 97.5% water content is provided. Supporting data and calculations are needed to substantiate this value, as it is central to the hydrogel's high water retention properties.

Response: Thanks for the suggestion. The water content is estimated based on the original mass of PAA240k-AM hydrogel and weight change from full swelling. The original polymer content of PAA240k-AM is 40 wt% regardless of the mass of initiator. Assume all monomers are fully polymerized, the weight change after swelling is attributed to water swelled. The water content of PAA240k-AM after swelling is:

$$\text{water content} = \frac{m_i \cdot 0.4}{(m_i \cdot 0.6 + (m_s - m_i))} \times 100\% \quad (3)$$

where m_i is the initial mass of hydrogel and m_s is the mass of swelled hydrogel. The detailed data is listed in table R8.

Hydrogel	Initial mass, m_i	Swelled mass, m_s	Water content
PAA240k-AM	0.395	4.210	96.1%

The water content of swelled PAA-AM is 96.1 wt% but not 97.5 wt%. We apologize for the miscalculation, and the detailed calculation is supplemented in supplementary information.

Revision: and the water content after swelling is 96.1 wt%. (Highlighted sentence in the sixteenth paragraph of Result and Discussion)

The water content is estimated based on the original mass of PAA240k-AM hydrogel and weight change from full swelling. The original polymer content of PAA240k-AM

is 40 wt% regardless of the mass of initiator. Assume all monomers are fully polymerized, the weight change after swelling is attributed to water swelled. The water content of PAA240k-AM after swelling is:

$$\text{water content} = \frac{m_i * 0.4}{(m_i * 0.6 + (m_s - m_i))} \times 100\%$$

where m_i is the initial mass of hydrogel and m_s is the mass of swelled hydrogel.

(Highlighted sentence in the Determination of water content in swelled hydrogel paragraph of Experimental procedures)

9. It is challenging to distinguish which stress-strech curves were used to calculate the residual strain% values presented. The authors are recommended to identify which data was used for this calculation.

Response: Thanks for the careful reading. Residual strains are recorded for all stress-stretch curves of cyclic loading. The residual strain is determined from the point where the slope of unloading curve abruptly changes because this change of slope indicates hydrogels bend instead of retracting. When hydrogels bend during uniaxial unloading, the polymer network experiences unrecoverable elongation, which is plastic deformation. Therefore, the residual strain is the indication of plastic deformation and reflects the elasticity of polymer network. We apologize for the typographical error where the initial residual strain is the first cycle but not the second cycle, and related clarification is added in the manuscript.

Revision: The initial residual strain of PAA240k-AM after the first cycle is 0.83% (Figure S18) the lowest among all PAA-AM hydrogels. (Highlighted sentence in the fifteenth paragraph of Result and Discussion)

The residual strains of all cyclic loading curves are tracked, (Highlighted sentence in the fifteenth paragraph of Result and Discussion)

10. The difference in color between samples in Figure 2a is difficult to discern. The authors are recommended to adjust the color scheme for greater contrast, ensuring each sample is easily distinguishable for readability.

Response: Thanks for the advice. The data density is reduced to make the fit line more obvious (Figure R18 and R19)

Figure R18. (A) SAXS profiles of PAA240k, PAA240k+AM, PAA3k and PAA3k+AM precursor solution with Lorentzian peak approximation. (B) SAXS profiles of PAA-AM hydrogels with correlation length approximation and Lorentzian peak approximation.

Figure R19. SAXS profiles of PAA240k-AM before and after swelling with Lorentzian peak approximation.

Revision: The corresponding change has been conducted in Figure 2A, 2C and 4C.

11. There are multiple typographical errors (e.g., in Figure S7, Figure S17ab). Please review and edit for correctness.

Response: Thanks for the careful reading. The corresponding typographical errors were reviewed and corrected.

Revision:

All caption titles in supplementary information are changed to 'Figure'

The corresponding typographical errors in Figure S4, Figure S7, Figure S16 and Figure S17 are corrected.